# Cross City Traffic Flow Generation via Retrieval Augmented Diffusion Model

**Yudong Li**[1], **Jingyuan Wang**[*1, 2, 3], **Xie Yu**[1], **Peiyu Wang**[1], **Qian Huang**[4]

[1]School of Computer Science and Engineering, Beihang University, Beijing, China
[2]School of Economics and Management, Beihang University, Beijing, China
[3]MIIT Key Laboratory of Data Intelligence and Management, Beihang University, Beijing, China
[4]Global Technical Service Dept, Huawei Technologies Co., Ltd Beijing, China
{yudongli, yjwang, yuxie_scse, peiyuwang}@buaa.edu.cn
huangqian16@huaiwei.com

## Abstract

Traffic flow data are of great value in smart city applications. However, limited by data collection costs and privacy sensitivity, it is rather difficult to obtain large-scale traffic flow data. Therefore, various data generation methods have been proposed in the literature. Nevertheless, these methods often require data from a specific city for training and are difficult to directly apply to new cities lacking data. To address this problem, this paper proposes a retrieval-augmented diffusion generation model with geographic representation alignment. We use data from multiple source cities for training, extract consistent representations across multiple cities, and leverage retrieval-augmented generation (RAG) technology to incorporate dynamic traffic flow patterns into the condition, aiming to improve the accuracy of data generation in the target city. Experiments on four real-world datasets demonstrate that, compared to existing generation methods, our method achieves best cross-city zero-shot performance. Our code and datasets can be found in `https://github.com/lyd1881310/CRAFT`.

## 1 Introduction

**Background.** Traffic flow data is crucial in intelligent transportation systems [26, 17], urban management [16, 49] and smart cities applications [11, 47]. The success of traffic flow analysis [48, 55] is coupled with a data-hungry paradigm, where superior performance and remarkable generalization ability rely on large-scale and high-quality data. However, unlike computer vision (CV) and natural language processing (NLP), where large-scale datasets are readily available from public sources, collecting traffic flow data faces strict privacy constraints and much higher costs. As a result, existing public traffic flow datasets are usually limited in both scale and quality. Overcoming this challenge to acquire larger-scale, higher-quality traffic flow data has emerged as a critical bottleneck.

**Motivation of This Work.** In response, traffic flow generation has become an increasingly urgent research direction [58]. It aims to synthesize realistic traffic flow by learning the conditional mapping from static urban geographic context to dynamic traffic flow distributions. Early work predominantly employed physics-driven models [60, 52, 1, 42, 28], which only estimate static average flow volumes and struggled to capture the complex temporal dynamics observed in real-world traffic. More recently, deep generative approaches have been dominant, leveraging richer geographic context as input to model flow distributions with greater fidelity. Their modeling paradigm has shifted from variational

---

[*]Corresponding author: Jingyuan Wang

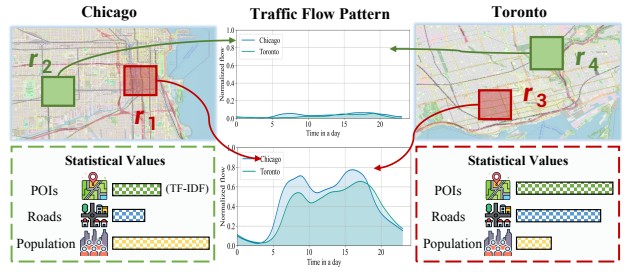 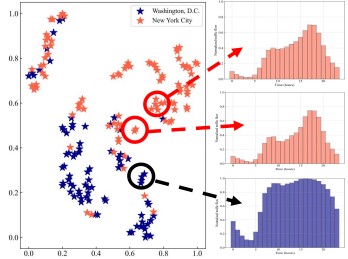

(a) There still exits correspondences between geographic contexts also and flow patterns even in cross-city scenarios

(b) Domain shift disrupts original correspondences

Figure 1: In 1a, For each region, we aggregate the statistical values of its geographic contexts to form the raw geographic features, finding that regions with similar geographic features may share flow patterns even across different cities. From 1b, we further observe that, overall, this correspondence becomes blurred in cross-city scenarios. Although geographic features and traffic flow are still roughly correlated, the presence of cross-city domain shift prevents it from being applied directly.

autoencoders (VAEs) [22] and generative adversarial networks (GANs) [8] to diffusion models [12], which now lead the field due to their strong conditional generation capabilities. Consequently, recent studies emphasize incorporating stronger urban priors into conditions [58], thereby supporting more fine-grained and accurate traffic flow generation.

*While incorporating stronger geographic priors into model inputs boost performance, these priors are often city-specific, which limits the model's ability to generalize.* Such priors are essentially the dependencies between local geographic structures and their corresponding flow patterns, which typically require training on city-specific historical flow data. This reliance hinders model deployment in cities with insufficient traffic flow records. *In this paper, we attempt to address this issue, and explore directly generating dynamic traffic flow in entirely unseen cities via* **a cross-city transfer learning framework.**

**Problem Analysis.** To achieve our motivation, we must restrict model learning from common geographic contexts shared across cities, thus avoiding the aforementioned city-specific priors.

In specific, they denote points of interest (POIs), roads, and population. However, utilizing these contexts for cross-city transfer faces two difficulties:

- *Domain Shift:* Within a single city, regions with similar geographic features typically exhibit analogous traffic-flow patterns. However, in cross-city settings, domain shifts in geographic representations disrupt this correspondence (as illustrated in Fig. 1). Consequently, regions from different cities—even if their geographic features are close in representation space—may exhibit markedly different traffic-flow behaviors (see in Fig. 1b).

- *Insufficient Condition:* static geographic contexts alone cannot support accurate cross-city flow generation. Regions with similar spatial characteristics may share similar periodicity and trends but their stochastic properties can vary significantly (e.g., absolute values of mean, peak and variance). Such dynamics are provided by historical flow records but are absent in cross-city scenarios.

*Remark:* Unlike traffic flow prediction [18, 50, 15, 46], which forecasts future traffic based on historical flow data, traffic flow generation synthesizes realistic traffic flow mainly on static geographic features. Departing from previous methods, our model is explicitly designed for cross-city generalization: it is trained on source cities and directly deployed to unseen target cities.

**Design Insights and Contributions.** Accordingly, we propose a Cross-city Retrieval-Augmented traffic Flow generaTion model (CRAFT), which is a simple yet effective DDPM-based [12] model. Fundamentally, we propose the Geographic Feature Alignment (GFA) to address *domain shift*; For *Insufficient Condition:*, we propose the Retrieval-based Condition Augmentation (RCA), which integrates traffic flow from source cities to supplement temporal dynamics. Our contributions can be summarized as:

- Our work is an initial probe to tackle cross-city traffic flow generation, proposing a transfer learning framework that enables traffic flow generation without historical data in target cities.

- Specifically, we proposed GFA and RCA to tackle two fundamental challenges in cross-city flow generation. Both are lightweight, plugin components without altering the backbone architecture, revealing the simplicity of our approach.

- Extensive experiments on four real-world urban datasets demonstrate the state-of-the-art (SOTA) zero-shot generation performance of our model and further validate its strong generalization ability.

## 2   Related Work

**Traffic Flow Generation Models** has evolved along two key trajectories: leveraging more comprehensive geographic features as inputs, and evolving from static to dynamic flow generation. This development can be grouped into three stages: physics-based, static, and dynamic flow generation models. *Physics-based Models* rely on empirical rules. The Gravity Models [1, 52, 60] and Radiation Models [28, 42] generate origin-destination (OD) flows based on gravitational laws and radiative diffusion, respectively. These methods use simple geographic features as inputs and are limited to coarse-grained OD flow prediction; Subsequently, neural networks became prevalent due to their ability to incorporate more comprehensive geographic information. Initially, *Static Flow Generation Models* learn the dependencies between static traffic flows and the geographic characteristics of urban regions [27, 40, 34, 37, 43]. For instance, DeepGravity [43] extracts features from OpenStreetMap (OSM) to estimate OD flows, while DeepFlowGen [40] incorporates abundant POI data. Although these methods capture richer geo-features than physics-based models, they still fall short in modeling temporal dynamics; Further, *Dynamic Flow Generation Models* address this limitation by modeling temporal variations through generative frameworks [58, 14, 56, 3, 36, 33, 35]. The field has advanced from generative adversarial networks (GANs) [14] to diffusion models [3, 58], which now lead the field due to their strong conditional generation capabilities.

*Comparison:* Current models perform well within individual cities as they tend to involve city-specific priors, but this reliance hinders their cross-city transfer abilities. In contrast, our model is designed to enable cross-city generalization by leveraging the common features shared across cities.

**Retrieval-Augmented Generation (RAG)** is widely used in large language models (LLMs) [24, 7], as it can dynamically integrate knowledge from external databases and enhance generation accuracy. Recently, time series analysis involves RAG to provide meaningful guidance [25]. For example, [25, 30, 44, 20] applied similarity retrieval based on time series embeddings to improve the prediction accuracy and enhance the zero-shot capabilities of time series foundation models (TSFM) [5, 54]. RAG is helpful for enhancing the model's capabilities in unseen scenarios. However, in the field of spatio-temporal data generation [21, 14, 59, 51], attempts using RAG still remain limited.

## 3   Preliminaries

**Definition 1: (Region)** We divided each city into $N$ non-overlapping basic rectangular grids or polygons, denoted as $\mathcal{R} = \{r_i | i = 1, 2, ..., N\}$, where $N$ is the total number of regions. Each $r_i$ is characterized by various geographical features including population, road network and points of interest (POIs). These features serve as the basic static condition for traffic flow generation.

**Definition 2: (Region Graph)** A region graph is denoted as $\mathcal{G} = \langle \mathcal{R}, \mathcal{E} \rangle$, where $\mathcal{E} \in \{0, 1\}^{|N| \times |N|}$ is the binary adjacency matrix. Specifically, $\mathcal{E}_{i,j} = 1$ if $r_i$ and $r_j$ are adjacent, and $\mathcal{E}_{i,j} = 0$ otherwise.

**Definition 3: (Traffic Flow)** Given a region $r_i$, assuming it has $C$ traffic flow features, such as traffic inflow and outflow. We denote the traffic flow of $r_i$ at $t$-th time slice with time length $T$ as $\boldsymbol{X}_{i,t} \in \mathbb{R}^{C \times T}$. The traffic flow dataset of all regions staring at all time slice is denoted as $\mathcal{X}$.

**Problem Statement: (Traffic flow generation)**. The objective of traffic flow generation is to train a model $\mathcal{F}$ to generate dynamic traffic flow of regions based on their static geographic features, including population, road network and POIs. Specifically, $\mathcal{F}$ takes a region graph $\mathcal{G}$ as input and outputs $\hat{\mathcal{X}}$ as

$$\hat{\mathcal{X}} = \mathcal{F}(\mathcal{G}; \theta), \tag{1}$$

where $\theta$ are trainable parameters of $\mathcal{F}$, and $\hat{\mathcal{X}}$ is evaluated by measuring the similarity to the real-world traffic flow data $\mathcal{X}$.

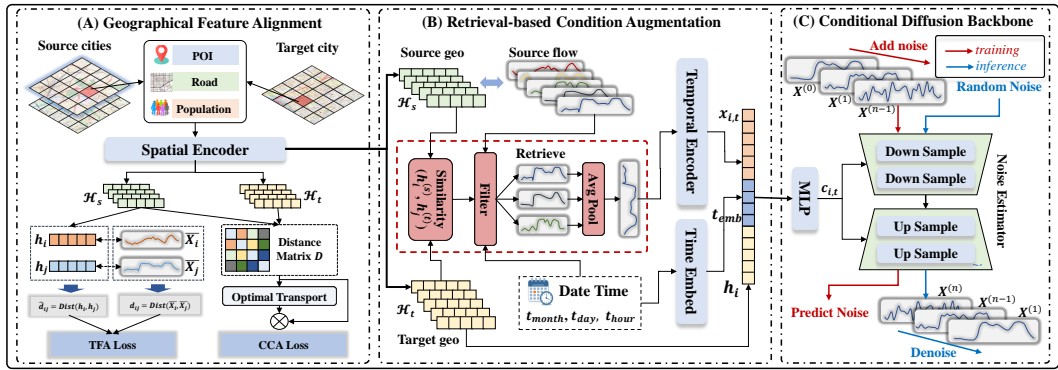

Figure 2: The framework of CRAFT

**Problem Statement: (Cross-city traffic flow generation)**. Models are trained on data from multiple source cities and make zero-shot generation on unseen target cities. Formally, let $\mathcal{G}^{(t)}$ denote the region graph of a target city and $\theta^{(s)}$ denote parameters trained from source cities and the process is

$$\hat{\mathcal{X}}^{(t)} = \mathcal{F}\left(\mathcal{G}^{(t)}; \theta^{(s)}\right). \tag{2}$$

## 4 Methodology

### 4.1 Framework

The framework of CRAFT is in Figure. 2, which applies the DDPM as the backbone. Specifically, CRAFT contains three modules: (1) Geographical Feature Alignment (GFA); (2) Retrieval-based Condition Augmentation (RCA); (3) Conditional Diffusion Backbone.

Initially, the datasets are divided into source and target cities: source cities include both traffic flow data and regional geographic features, while the target city only has regional geographic features. GFA is first pre-trained on source and target cities to provide cross-city geographic feature alignment. Subsequently, RCA is designed to retrieve relevant historical data from source cities via the aligned regional representations, supplying an augmented condition for the diffusion backbone. Finally, Conditional Diffusion Backbone takes augmented condition to make zero-shot traffic flow generation in the target city.

### 4.2 Geographic Feature Alignment

To address the challenge of *domain shift* (in Section 1), we propose the geographic feature alignment module. Unlike existing geographic representation learning methods [19, 4, 9, 57, 53], which learn representations based on entity embeddings, we begin by selecting common geographic features shared across cities and use them to construct basic representations for each region. Subsequently, we adopt the Graph Transformer [41] as the spatial encoder, which takes each city's region graph $\mathcal{G}$ as input to generate higher-level representations. Finally, the encoder is trained through traffic flow alignment (TFA) and cross-city alignment (CCA) to enable transferable geographic representations.

**Basic Geographic Representation.** For most cities, both static maps (OpenStreetMap) and population data (WorldPop) are publicly available and highly correlated with traffic flow patterns. Therefore, the basic geographic representations are derived from the following three aspects:

- *POIs:* POIs are highly correlated with the region functions. For each region, we use the TF-IDF algorithm [39] to construct POI representations $\boldsymbol{f}^{(poi)}$ based on POIs' categories and numbers.

- *Roads:* For each region, we compute the total length of road segments across all road categories, as this captures the region's transportation capacity. Regions with dense networks and a high proportion of trunk roads typically offer better accessibility and attract heavier volume of traffic flow. The road feature is denoted as $\boldsymbol{f}^{(road)}$.

- *Population:* For each region, we collect statistics on its population size and density, as population implicitly reflect the scale of its traffic flow. Regions with higher population density tend to generate larger volumes of traffic flow. The population feature is denoted as $\boldsymbol{f}^{(pop)}$.

For each region $r_i$, its $\boldsymbol{f}^{(poi)}$, $\boldsymbol{f}^{(road)}$, and $\boldsymbol{f}^{(pop)}$ are concatenated and projected through a multi-layer perceptron (MLP) to generate the *basic geographic representation*

$$\boldsymbol{z}_i = \mathbf{MLP}\left(\boldsymbol{f}_i^{(pop)} \| \boldsymbol{f}_i^{(poi)} \| \boldsymbol{f}_i^{(road)}\right). \tag{3}$$

Further, the spatial encoder takes $\boldsymbol{z}_i$ as input and models the correlations among regions. For a specific city with its $\mathcal{G}$. Region representations are further embedded as

$$\{\boldsymbol{h}_1, \boldsymbol{h}_2, ..., \boldsymbol{h}_N\} = \text{GraphTransformer}\left(\{\boldsymbol{z}_1, \boldsymbol{z}_2, ..., \boldsymbol{z}_N\}, \mathcal{G}\right), \tag{4}$$

where $N$ is the region numbers of this city and $\mathcal{G}$ is the region graph. To obtain cross-city transferable region representations, we pre-train the spatial encoder through the proposed traffic flow alignment and cross-city alignment methods. During the pre-training stage, geographic representations of both source and target cities are jointly involved to facilitate cross-city alignment.

**Traffic Flow Alignment (TFA)**. Only regions from source cities are involved in the TFA. We aim to ensure that the differences in region representations reflect the differences in their corresponding traffic flows. Specifically, we first compute the average traffic flow for each region over the time period. Then, all regions from the source cities are collected as the set $\mathcal{R}_s = \{r_1, r_2, \cdots, r_{N_s}\}$, with their corresponding representations denoted as $\mathcal{H}_s = \left\{h_1^{(s)}, h_2^{(s)}, \cdots, h_{N_s}^{(s)}\right\}$, where $N_s$ is the total number of regions from all source cities. The traffic flow alignment loss is defined as

$$\mathcal{L}_{FA} = \frac{1}{N_s^2}\sum_{i,j}\left(\hat{d}_{ij} - d_{ij}\right)^2, \tag{5}$$

where $\hat{d}_{ij} = \text{MinMaxNorm}_{i,j}\left(\left\|\boldsymbol{h}_i^{(s)} - \boldsymbol{h}_j^{(s)}\right\|_2\right)$, $d_{ij} = \text{MinMaxNorm}_{i,j}\left(\left\|\bar{\boldsymbol{X}}_i - \bar{\boldsymbol{X}}_j\right\|_2\right)$, and $\bar{\boldsymbol{X}}_i$ is the average traffic flow data of region $r_i$.

**Cross-City Alignment (CCA)**. We seek to project semantically similar regions from both source and target cities into proximity within the representation space. Since explicit regional correspondence labels are unavailable, our model needs to learn these correspondences adaptively. By leveraging flow patterns, TFA alone draws regions with similar flows together. Although it does not fully resolve domain shift, it groups regions with both similar flow dynamics and geographic context. Consequently, in the representation space each region's counterparts naturally reside in its neighborhood. Therefore, we formulate the cross-city alignment as an optimal transport (OT) problem [32], seeking the minimum-cost distribution mapping between regions in the source and target cities under the Wasserstein distance.

Specifically, the representations of all regions in source cities and target city are collected into the set $\mathcal{H}_s$ and $\mathcal{H}_t = \left\{h_1^{(t)}, h_2^{(t)}, \cdots, h_{N_t}^{(t)}\right\}$, respectively. $N_t$ is the number of regions in the target city. As both $\mathcal{H}_s$ and $\mathcal{H}_t$ are enumerable, we can directly calculate the Wasserstein distance and tackle this problem through the optimal transport solver [2, 6]. First, Euclidean distance is used to get the initial representation distance matrix $\boldsymbol{D}$ between regions in source cities and the target city, and $\boldsymbol{D}_{ij}$ is

$$\boldsymbol{D}_{ij} = \left\|\boldsymbol{h}_i^{(s)} - \boldsymbol{h}_j^{(t)}\right\|_2, \text{where } \boldsymbol{h}_i^{(s)} \in \mathcal{H}_s, \boldsymbol{h}_j^{(t)} \in \mathcal{H}_t. \tag{6}$$

Then, the transport matrix $\boldsymbol{T}$ is calculated by the OT solver, where $\boldsymbol{T} = \text{OTSolver}\left(\boldsymbol{D}\right)$. The cross-city alignment loss $\mathcal{L}_{CA}$ is defined as Wasserstein distance, which is equal to the dot product of the transport matrix $\boldsymbol{T}$ and the distance matrix $\boldsymbol{D}$, as follows

$$\mathcal{L}_{CA} = \sum_{ij}\boldsymbol{T}_{ij} \cdot \boldsymbol{D}_{ij}. \tag{7}$$

For more details on the formulation of optimal transport problem and the structure of the OT solver, please refer to Appendix A. The total alignment loss is the weighted sum of $\mathcal{L}_{FA}$ and $\mathcal{L}_{CA}$, where $\lambda_1$ and $\lambda_2$ are the balance weights, as follows

$$\mathcal{L}_A = \lambda_1\mathcal{L}_{FA} + \lambda_2\mathcal{L}_{CA}. \tag{8}$$

*Remark:* As shown in Fig. 1, basic geographic representations are too coarse to guide cross-city flow generation. To refine this issue, we propose GFA with two complementary losses: **(1)** Traffic-flow alignment: refine regional representations in the correct direction according to regional flow patterns from source cities; **(2)** Tackle cross-city alignment without explicit regional correspondence labels, it enables our model to adaptively determine the most accurate counterpart by leveraging an optimal transport loss.

### 4.3  Retrieval-based Condition Augmentation

To address the challenge of *insufficient condition* (Section 1), we proposed the retrieval-augmented condition augmentation strategy. For each target region, our method retrieves relevant historical data from source cities according to its geographic representation, supplementing the input conditions for the diffusion backbone. For each target region $r_i$, the generated input condition contains three parts: geographic features $\boldsymbol{h}_i$, time embedding $\boldsymbol{t}_{emb}$, and historical data from source cities $\boldsymbol{x}_{i,t}$.

$\boldsymbol{t}_{emb}$ contains three periodic temporal components: the month in the year $t_{month} \in [1, 12]$, the day in the week $t_{day} \in [1, 7]$ and the hour in the day $t_{hour} \in [1, 24]$. These three components are encoded into vectors through the embedding layers and concatenated to obtain the time embedding $\boldsymbol{t}_{emb} = (\boldsymbol{t}_{month} \| \boldsymbol{t}_{day} \| \boldsymbol{t}_{hour})$, which indicates the start time of the generated flow.

$\boldsymbol{x}_{i,t}$ denotes the retrieved results for $r_i$ according to $t_{month}, t_{day}, t_{hour}$ and $\boldsymbol{h}_i$. Traffic flows from source cities are first segmented into sequences of length $T$. We then filter these segments using two criteria: (1) Starting time information $t_{month}, t_{day}$ and $t_{hour}$; (2) Region representation similarity to $\boldsymbol{h}_i$. The matched top $K$ flow sequences are selected as $\bar{\mathcal{X}}_{i,t} = \{\bar{\boldsymbol{X}}_{1,t}, \bar{\boldsymbol{X}}_{2,t}, ..., \bar{\boldsymbol{X}}_{K,t}\}$, where $\bar{\boldsymbol{X}}_{k,t} \in \mathbb{R}^{C \times T}$, where $\bar{\boldsymbol{X}}_{i,t}$ is the average flow of region $r_i$ staring at time slice $t$. We employ a self-attention [45] block $\mathbf{Attn}(\cdot)$ to extract flow patterns from $\bar{\mathcal{X}}_{i,t}$, which takes the averaged $\bar{\mathcal{X}}_{i,t}$ as input to mitigate the impact of noise. The extracted retrieval result is

$$\boldsymbol{x}_{i,t} = \mathbf{Attn}\left(\frac{1}{K}\sum_{k=1}^{K}\bar{\boldsymbol{X}}_{k,t}\right). \tag{9}$$

Finally, above features are concatenated and projected by a Multi-Layer Perceptron (MLP) block to generate the input condition $\boldsymbol{c}_{i,t}$ for the target region $r_i$, as follows

$$\boldsymbol{c}_{i,t} = \mathbf{MLP}\left(\boldsymbol{h}_i \| \boldsymbol{x}_{i,t} \| \boldsymbol{t}_{emb}\right). \tag{10}$$

### 4.4  Conditional Diffusion Backbone

As shown in Fig. 2, we adopt the Denoising Diffusion Probabilistic Model (DDPM) as the backbone.

**Training.** In the training phase, for the traffic flow sample $\boldsymbol{X}_{i,t}$ in source city, we add random noise through the forward diffusion process and use a 1D-U-Net [38] as a noise estimator to predict the added noise. Following most of the literature about diffusion model, we use $\boldsymbol{X}$ to represent $\boldsymbol{X}_{i,t}$ in the rest of this paper. Following [12], the noise-adding process of $k$ steps can be simplified as:

$$\boldsymbol{X}^{(k)} = \sqrt{\bar{\alpha}_k}\boldsymbol{X} + \sqrt{1 - \bar{\alpha}_k}\boldsymbol{\epsilon}, \tag{11}$$

where $\boldsymbol{\epsilon} \sim \mathcal{N}(\mathbf{0}, \boldsymbol{I})$ and $\bar{\alpha}_k$ is a hyper parameter. The noise estimator $\boldsymbol{\epsilon}_\theta$ takes the noised sample $\boldsymbol{X}^{(k)}$, the step $k$ and the condition vector $\boldsymbol{c}_{i,t}$ as inputs to predict the noise. The Mean Squared Error (MSE) loss function is used to train the noise estimator, as follows

$$\mathcal{L}_{NE} = \mathbb{E}_{k, \boldsymbol{X}_{i,t}, \boldsymbol{\epsilon}} \left\| \boldsymbol{\epsilon}_\theta\left(\boldsymbol{X}^{(k)}, k, \boldsymbol{c}_{i,t}\right) - \boldsymbol{\epsilon} \right\|^2. \tag{12}$$

**Inference.** In the inference phase, for each $r_i$ in the target city, the model takes a random Gaussian noise $\boldsymbol{\epsilon} \sim \mathcal{N}(\mathbf{0}, \boldsymbol{I})$ as input and generates its corresponding traffic flow $\hat{\boldsymbol{X}}$ conditioned on the $\boldsymbol{c}_{i,t}$ through an $n$-step denoising process:

$$p_\theta\left(\hat{\boldsymbol{X}}^{(k-1)} | \hat{\boldsymbol{X}}^{(k)}\right) := \mathcal{N}\left(\boldsymbol{\mu}_\theta\left(\hat{\boldsymbol{X}}^{(k)}, k, \boldsymbol{c}_{i,t}\right), \boldsymbol{\sigma}^2\left(\hat{\boldsymbol{X}}^{(k)}, k\right)\boldsymbol{I}\right), \ 1 \leq k \leq n, \tag{13}$$

In the equation 13, $\boldsymbol{\mu}_\theta$ denotes the mean variable and $\boldsymbol{\sigma}^2$ is the variance:

$$\boldsymbol{\mu}_\theta\left(\boldsymbol{X}^{(k)}, k, \boldsymbol{c}_{i,t}\right) = \frac{1}{\sqrt{\bar{\alpha}_k}}\left(\boldsymbol{X}^{(k)} - \frac{\beta_k}{\sqrt{1 - \bar{\alpha}_k}}\boldsymbol{\epsilon}_\theta\left(\boldsymbol{X}^{(k)}, k, \boldsymbol{c}_{i,t}\right)\right); \ \boldsymbol{\sigma}^2\left(\boldsymbol{X}^{(k)}, k\right) = \frac{1 - \bar{\alpha}_{k-1}}{1 - \bar{\alpha}_k}\beta_k. \tag{14}$$

Table 1: Cross-city traffic flow generation results

| Method | City | Inflow | | | Outflow | | | City | Inflow | | | Outflow | | |
|---|---|---|---|---|---|---|---|---|---|---|---|---|---|---|---|
| | | CPC(↑) | NMAE(↓) | NRMSE(↓) | CPC(↑) | NMAE(↓) | NRMSE(↓) | | CPC(↑) | NMAE(↓) | NRMSE(↓) | CPC(↑) | NMAE(↓) | NRMSE(↓) |
| GMEL | Chicago(CHI) | 0.730 | 0.173 | 0.227 | 0.725 | 0.175 | 0.236 | Washington,D.C.(DC) | 0.741 | 0.273 | 0.319 | 0.716 | 0.281 | 0.316 |
| DFG | | 0.162 | 0.306 | 0.438 | 0.159 | 0.310 | 0.441 | | 0.691 | 0.240 | 0.343 | 0.690 | 0.239 | 0.342 |
| KSTDiff | | 0.006 | 0.334 | 0.467 | 0.132 | 0.322 | 0.453 | | 0.613 | 0.519 | 0.610 | 0.608 | 0.564 | 0.673 |
| CGAN | | 0.230 | 0.379 | 0.511 | 0.203 | 0.364 | 0.495 | | 0.514 | 0.384 | 0.509 | 0.507 | 0.398 | 0.523 |
| Diffwave | | 0.332 | 0.486 | 0.598 | 0.532 | 0.496 | 0.609 | | 0.650 | 0.403 | 0.538 | 0.570 | 0.458 | 0.584 |
| DiT | | 0.509 | 0.357 | 0.451 | 0.528 | 0.343 | 0.436 | | 0.634 | 0.351 | 0.420 | 0.607 | 0.352 | 0.422 |
| DDPM | | 0.415 | 0.273 | 0.393 | 0.416 | 0.275 | 0.396 | | 0.346 | 0.387 | 0.514 | 0.351 | 0.388 | 0.512 |
| CVAE | | 0.490 | 0.267 | 0.385 | 0.488 | 0.269 | 0.385 | | 0.468 | 0.343 | 0.461 | 0.471 | 0.347 | 0.461 |
| **CRAFT** | | **0.785** | **0.140** | **0.216** | **0.786** | **0.140** | **0.216** | | **0.815** | **0.158** | **0.240** | **0.816** | **0.159** | **0.240** |
| GMEL | Toronto(TRT) | 0.735 | 0.224 | 0.283 | 0.744 | 0.217 | 0.276 | New York City(NYC) | 0.585 | 0.310 | 0.379 | 0.672 | 0.188 | 0.242 |
| DFG | | 0.278 | 0.395 | 0.514 | 0.278 | 0.394 | 0.512 | | 0.581 | 0.182 | 0.278 | 0.585 | 0.183 | 0.279 |
| KSTDiff | | 0.006 | 0.469 | 0.597 | 0.248 | 0.413 | 0.540 | | 0.040 | 0.226 | 0.364 | 0.030 | 0.228 | 0.364 |
| CGAN | | 0.561 | 0.358 | 0.468 | 0.572 | 0.354 | 0.461 | | 0.368 | 0.330 | 0.476 | 0.407 | 0.382 | 0.527 |
| Diffwave | | 0.521 | 0.421 | 0.531 | 0.546 | 0.456 | 0.574 | | 0.363 | 0.475 | 0.608 | 0.434 | 0.440 | 0.575 |
| DiT | | 0.552 | 0.394 | 0.489 | 0.601 | 0.381 | 0.471 | | 0.419 | 0.416 | 0.508 | 0.449 | 0.383 | 0.478 |
| DDPM | | 0.592 | 0.358 | 0.474 | 0.595 | 0.359 | 0.474 | | 0.523 | 0.316 | 0.431 | 0.525 | 0.324 | 0.438 |
| CVAE | | 0.699 | 0.289 | 0.394 | 0.695 | 0.295 | 0.398 | | 0.568 | 0.280 | 0.403 | 0.571 | 0.280 | 0.401 |
| **CRAFT** | | **0.804** | **0.178** | **0.267** | **0.804** | **0.179** | **0.268** | | **0.782** | **0.103** | **0.170** | **0.786** | **0.102** | **0.165** |

where $\beta_k$ is a hyper-parameter and $\bar{\alpha}_k = \prod_{k=1}^{k} (1 - \beta_k)$. Through an n-step denoising process, the traffic flow data $\hat{X}^{(0)}$ is generated.

# 5   Experiment

*Dataset:* We conducted experiments on four real-world bicycle trip datasets, namely Chicago (CHI)[1], Washington, D.C. (DC)[2], Toronto (TRT) [3], and New York City (NYC) [4]. We manually partition each city into grid-based regions and count the number of bicycles entering and exiting each region within each hour to obtain the traffic flow. In our experiments, we use normalized traffic flow values; *Baseline:* We employ GMEL [27], DFG [40], KSTDiff [58], CGAN [29], Diffwave [23], DiT [31], DDPM [12] and CVAE [10] as our baselines; *Evaluation Metric:* We evaluate our model on the metrics of Common Part of Commuters (CPC), Normalized Mean Absolute Error (NMAE) and Normalized Root Mean Square Error (NRMSE). *Details about dataset pre-procession, baselines, and evaluation metrics are in Appendix B.*

## 5.1   Overall Generation Performance

**Generation Performance:** Table. 1 reports the zero-shot cross-city performance comparison between our method and selected baselines. The experiments are organized into four groups. Each city is sequentially assigned as the target city while the remaining three cities serve as source cities. Models are trained exclusively on the source cities and evaluated in a zero-shot manner on the unseen target city. For each target city, both inflow and outflow data are generated. From the results, our method consistently outperforms existing approaches and achieves state-of-the-art (SOTA) performance across various settings. It yields an improvement of 59.7% compared with the average level of all baselines and an improvement of 22.5% compared with the second-best baseline (GMEL). When compared with ordinary DDPM, our approach yields an average improvement of 61.5%.

**Utility of the Generated Data:** To further assess the quality of the generated flow data, we evaluate them on a downstream traffic flow task across multiple target cities. Specifically, we use the synthetic data produced by CRAFT and other baselines to train two representative models for flow prediction: a vanilla LSTM [13] and a Transformer [45]. As a reference, we also train the same models on real flow data from the corresponding target cities. As shown in Table. 2, our model consistently outperforms all baseline methods across various evaluation metrics. It achieves an improvement of 55.9% over the average level of all baselines and an improvement of 14.9% over the second-best baseline (DDPM). Furthermore, our model achieves performance that is closest to the results of direct training on real traffic flow data. Specifically, it demonstrates an average performance degradation of only 10.4%, with a minimum drop of 3.8% and a maximum drop of 22.2%. These results underscore the model's

---

[1] https://divvy-tripdata.s3.amazonaws.com

[2] https://s3.amazonaws.com/capitalbikeshare-data

[3] https://ckan0.cf.opendata.inter.prod-toronto.ca

[4] https://s3.amazonaws.com/tripdata/2023-citibike-tripdata.zip

Table 2: Data utility comparison on traffic flow prediction

| Gen | Pred | Chicago(CHI) | | | | Washington, D.C.(DC) | | | | Toronto(TRT) | | | | New York City(NYC) | | | |
|---|---|---|---|---|---|---|---|---|---|---|---|---|---|---|---|---|---|
| | | Inflow | | Outflow | | Inflow | | Outflow | | Inflow | | Outflow | | Inflow | | Outflow | |
| | | MAE | RMSE | MAE | RMSE | MAE | RMSE | MAE | RMSE | MAE | RMSE | MAE | RMSE | MAE | RMSE | MAE | RMSE |
| Real | | 0.102 | 0.152 | 0.105 | 0.157 | 0.105 | 0.156 | 0.109 | 0.163 | 0.132 | 0.191 | 0.132 | 0.191 | 0.063 | 0.107 | 0.063 | 0.106 |
| GMEL | LSTM | 0.198 | 0.264 | 0.195 | 0.262 | 0.242 | 0.325 | 0.239 | 0.303 | 0.265 | 0.329 | 0.258 | 0.321 | 0.144 | 0.216 | 0.140 | 0.201 |
| DFG | | 0.303 | 0.443 | 0.307 | 0.446 | 0.157 | 0.232 | 0.154 | 0.233 | 0.318 | 0.436 | 0.319 | 0.437 | 0.082 | 0.141 | 0.081 | 0.139 |
| KSTDiff | | 0.339 | 0.472 | 0.306 | 0.441 | 0.514 | 0.604 | 0.560 | 0.672 | 0.478 | 0.606 | 0.418 | 0.551 | 0.225 | 0.363 | 0.229 | 0.365 |
| CGAN | | 0.435 | 0.562 | 0.414 | 0.540 | 0.450 | 0.571 | 0.447 | 0.566 | 0.340 | 0.445 | 0.347 | 0.451 | 0.291 | 0.432 | 0.334 | 0.475 |
| Diffwave | | 0.275 | 0.383 | 0.277 | 0.376 | 0.321 | 0.418 | 0.347 | 0.442 | 0.373 | 0.445 | 0.363 | 0.437 | 0.211 | 0.306 | 0.177 | 0.278 |
| DiT | | 0.266 | 0.315 | 0.251 | 0.307 | 0.307 | 0.361 | 0.301 | 0.347 | 0.334 | 0.382 | 0.323 | 0.363 | 0.246 | 0.280 | 0.250 | 0.282 |
| DDPM | | 0.117 | 0.176 | 0.119 | 0.177 | 0.163 | 0.240 | 0.165 | 0.245 | 0.150 | 0.221 | 0.161 | 0.234 | 0.081 | 0.130 | 0.084 | 0.129 |
| CVAE | | 0.252 | 0.344 | 0.256 | 0.348 | 0.293 | 0.383 | 0.289 | 0.376 | 0.243 | 0.344 | 0.244 | 0.344 | 0.142 | 0.233 | 0.142 | 0.231 |
| **CRAFT** | | **0.109** | **0.164** | **0.111** | **0.166** | **0.124** | **0.180** | **0.130** | **0.189** | **0.141** | **0.203** | **0.145** | **0.206** | **0.067** | **0.112** | **0.069** | **0.110** |
| Real | | 0.098 | 0.151 | 0.100 | 0.154 | 0.095 | 0.149 | 0.100 | 0.157 | 0.131 | 0.196 | 0.132 | 0.197 | 0.056 | 0.103 | 0.056 | 0.103 |
| GMEL | Transformer | 0.194 | 0.268 | 0.189 | 0.263 | 0.229 | 0.297 | 0.232 | 0.295 | 0.256 | 0.325 | 0.256 | 0.327 | 0.141 | 0.202 | 0.127 | 0.185 |
| DFG | | 0.337 | 0.484 | 0.311 | 0.451 | 0.251 | 0.338 | 0.236 | 0.314 | 0.331 | 0.447 | 0.336 | 0.453 | 0.083 | 0.145 | 0.089 | 0.148 |
| KSTDiff | | 0.336 | 0.469 | 0.338 | 0.477 | 0.514 | 0.605 | 0.560 | 0.672 | 0.479 | 0.607 | 0.418 | 0.550 | 0.208 | 0.338 | 0.218 | 0.357 |
| CGAN | | 0.373 | 0.487 | 0.380 | 0.496 | 0.393 | 0.503 | 0.403 | 0.508 | 0.286 | 0.382 | 0.297 | 0.388 | 0.264 | 0.391 | 0.276 | 0.406 |
| Diffwave | | 0.282 | 0.384 | 0.261 | 0.351 | 0.318 | 0.396 | 0.303 | 0.382 | 0.362 | 0.432 | 0.366 | 0.436 | 0.178 | 0.271 | 0.175 | 0.258 |
| DiT | | 0.275 | 0.317 | 0.253 | 0.304 | 0.297 | 0.352 | 0.295 | 0.347 | 0.333 | 0.381 | 0.322 | 0.363 | 0.229 | 0.267 | 0.233 | 0.270 |
| DDPM | | 0.117 | 0.181 | 0.122 | 0.186 | 0.163 | 0.247 | 0.164 | 0.251 | 0.150 | 0.222 | 0.159 | 0.235 | 0.078 | 0.133 | 0.081 | 0.131 |
| CVAE | | 0.224 | 0.308 | 0.224 | 0.308 | 0.271 | 0.351 | 0.267 | 0.344 | 0.253 | 0.339 | 0.255 | 0.341 | 0.160 | 0.269 | 0.163 | 0.268 |
| **CRAFT** | | **0.105** | **0.161** | **0.108** | **0.165** | **0.116** | **0.171** | **0.122** | **0.182** | **0.141** | **0.206** | **0.144** | **0.207** | **0.065** | **0.111** | **0.067** | **0.111** |

(a) Inflow - CPC  (b) Inflow - NMAE  (c) Inflow - NRMSE

(d) Outflow - CPC  (e) Outflow - NMAE  (f) Outflow - NRMSE

Figure 3: The overall ablation results on metrics of CPC (↑), NMAE (↓), NRMSE (↓)

(a) TFA + CCA  (b) only CCA  (c) TFA + CCA  (d) only TFA

Figure 4: Alignment Analysis on TFA and CCA (New York City is the Target City). All scatter points in this figures represent the regions' geographic features. In (a) and (b), the color intensity indicates the value of traffic flow. After TFA, regions with similar flow patterns exhibit similar geographic representations. By comparing (c) and (d), we observe that CCA alleviates the distribution shift between the target city and source cities.

superior capability in traffic flow generation, outperforming existing baselines and highlighting its strong potential for practical deployment.

## 5.2 Model Analysis

**Ablation Study:** Recall that our CRAFT relies on two main components: GFA and RCA. In Fig. 3, we first conducted overall ablation studies to testify their effectiveness based on following settings. (1) *w/o Alignment*: remove GFA by directly using target region's original geographic features for cross-

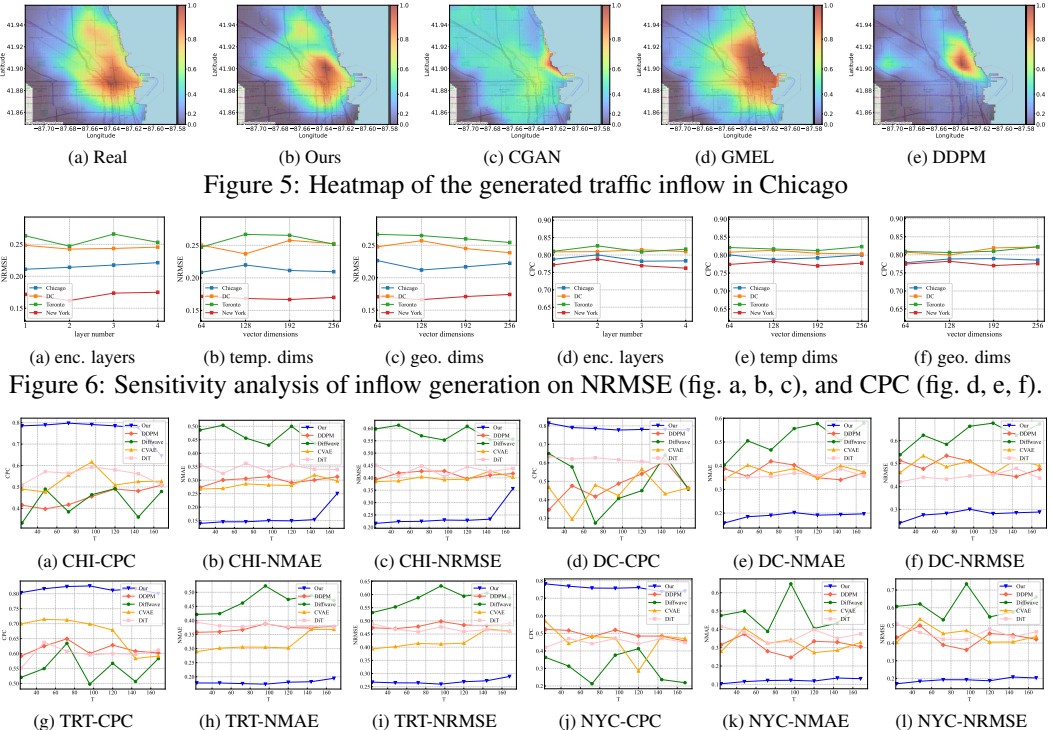

Figure 5: Heatmap of the generated traffic inflow in Chicago

(a) Real  (b) Ours  (c) CGAN  (d) GMEL  (e) DDPM

(a) enc. layers  (b) temp. dims  (c) geo. dims  (d) enc. layers  (e) temp dims  (f) geo. dims

Figure 6: Sensitivity analysis of inflow generation on NRMSE (fig. a, b, c), and CPC (fig. d, e, f).

(a) CHI-CPC  (b) CHI-NMAE  (c) CHI-NRMSE  (d) DC-CPC  (e) DC-NMAE  (f) DC-NRMSE

(g) TRT-CPC  (h) TRT-NMAE  (i) TRT-NRMSE  (j) NYC-CPC  (k) NYC-NMAE  (l) NYC-NRMSE

Figure 7: Temporal length extension of all four cities on metrics of CPC ($\uparrow$), NMAE ($\downarrow$), NRMSE ($\downarrow$). All metrics are evaluated on inflow generation.

city retrieval and generation of conditions. (2) *w/o RCA*: remove the retrieval-augmented features $x_{i,t}$ from $c_{i,t}$. (3) *w/o Temporal embedding*: remove the temporal embeddings $t_{emb}$ from $c_{i,t}$, which supplements the periodic patterns of the traffic flow. Both *w/o RCA* and *w/o Temporal embedding* are related to RCA. From the experimental results in Fig. 3 we can notice: (1) GFA provides the most improvement, highlighting that distribution shift is indeed the major problem in the cross-city generation. (2) In RCA, $t_{emb}$ contributes the most to the performance gain, highlighting the value of periodic temporal patterns. Additionally, $x_{i,t}$ consistently improves performance, highlighting the value of supplementing dynamic patterns from source cities.

Further, we explore the specific role of the traffic flow alignment (TFA) and the cross-city alignment (CCA) in GFA. In Fig. 4, we apply t-SNE to all regions' geographic features in New York. We observe that TFA indeed aligns geographic features with traffic flow patterns. CCA mitigates the distribution shift in geographic features between source cities and the target city, thereby enhancing cross-city generalization. Experiments on all four city are provided in Appendix C.

**Sensitivity Analysis:** We made various sensitivity analyses on four important hyperparameters, including (1) the layer number of the sequence encoder; (2) the dimension of temporal embedding; (3) the dimension of the geographic representation. As shown in Fig. 6, our model demonstrates strong robustness, with a maximum performance fluctuation of $8.8\%$. Detailed results for all evaluation metrics are provided in Appendix C.

### 5.3 Additional Experiments and Discussion

**Geographic Visualization:** Fig. 5 compares generated flow distributions of CRAFT and baselines in Chicago (please refer Appendix C for all cities). CRAFT and GMEL are most identical to the real distributions. Notably, CRAFT outperforms other baselines with more realistic generated flow. This highlights of CRAFT's strong zero-shot generation ability and better alignment with geographic and flow patterns.

**Temporal Length Extension:** In general, most models default to a horizon of $T = 24$, where most baselines remain stable. Increasing $T$ both raises task difficulty and enriches sample information, so an ideal model should exploit the enriched information from bigger $T$ while remaining stable in performance loss. To further explore the potential of each model's long horizon generation, we trained

several baselines and CRAFT on $\{2T, 3T, \cdots, 7T\}$ as $(48, 72, \cdots, 168)$. As Fig. 7 shows: (1) All models' performance drops as $T$ grows, revealing a common long-horizon bottleneck. (2) Diffwave exhibits high variance and poor accuracy, while diffusion-based DiT and DDPM remain steadier and more accurate. (3) CRAFT consistently achieves SOTA and outperforms all these baselines, demonstrating its superior potential in long-horizon generation. More results are in Appendix C.

## 6 Conclusion

In this paper, we propose CRAFT, a model capable of zero-shot traffic flow generation. Unlike existing approaches, CRAFT is specifically designed for cross-city scenarios and advantage in generating high-quality traffic flow data in cities with limited or no historical flow records. Specifically, we identify two key challenges: *domain shift* and *insufficient condition*. To address these, we introduce Geographic Feature Alignment (GFA) for domain shift and Retrieval-based Condition Augmentation (RCA) for insufficient condition. As both GFA and RCA serve as concise plug-and-lay modules, our method requires no additional modifications to the diffusion backbone. Extensive experiments on four real-world datasets demonstrate the broad effectiveness of CRAFT, and further illustrates the great potential of CRAFT in the fields of urban planning and traffic management.

**Limitations:** Restrict by available datasets, CRAFT is currently validated on in–out flow generation, and we have not yet explored its ability on other data types like origin–destination (OD) flows. Furthermore, restricted by computational resources, we do not conduct experiments on temporal horizon more than 168. Actually, from Fig. 7, CRAFT remains the most stable across varying temporal horizons, demonstrating its great potential for extension to longer temporal horizons.

## Acknowledgements

Jingyuan Wang's work was partially supported by the National Natural Science Foundation of China (No. 72222022, 72171013, 72242101) and the Fundamental Research Funds for the Central Universities (JKF-2025017226182).

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

# A   Model Details & Hyperparameters

## A.1   Basic Geographic Features

We extract urban geographic features from OpenStreetMap (OSM) [5] and WorldPop [6] data. The original geographic features include the following three aspects.

- POI features (denoted as $\boldsymbol{f}^{(poi)}$) reflect the function of a region. We attempt to construct the POI semantic features of a region by counting the quantities of various POI categories. However, the quantities of different POI categories exhibit significant imbalance, for instance, the total number of commercial POIs is far larger than that of residential POIs. This imbalance can easily cause the model to overlook the influence of POI categories with smaller quantities. Therefore, we employ the TF-IDF algorithm [39] to extract POI features. Specifically, we treat each POI category as a "word", consider all POIs within a region as a "document", and define the entire city as the "corpus". When calculating the importance of a word for a document, the algorithm automatically incorporates the word's frequency in the corpus for weighted analysis.

- Road features (denoted as $\boldsymbol{f}^{(road)}$) reflect the transportation attribute of a region. We calculate the total length of all categories of road segments within a region. Regions with dense road networks and a large number of trunk roads usually have convenient transportation and tend to generate a higher volume of traffic flow.

- Population features (denoted as $\boldsymbol{f}^{(pop)}$) reflect the traffic potential of a region. Regions with higher population density are more likely to generate a higher volume of traffic flow. We obtained the United Nations (UN)-adjusted $100m$ resolution national population data from *WorldPop* and counted the population number in each manually partitioned rectangular regions.

## A.2   Optimal Transport Problem

Optimal Transport (OT) is a mathematical problem aiming to find the most efficient way to move mass from source distribution to target distribution. It was introduced by *Gaspard Monge* in 1781. When both the source and target distributions are represented by enumerable samples ($N_s$ samples for source and $N_t$ samples for target), the OT problem can be formally defined as

$$\boldsymbol{T}^* = \arg\min_{\boldsymbol{T} \in \mathbb{R}_+^{N_s \times N_t}} \sum_{i,j} \boldsymbol{T}_{i,j} \cdot \boldsymbol{D}_{i,j},$$
$$s.t.\ \boldsymbol{T}\mathbf{1} = \boldsymbol{w}_s\ \text{ and }\ \boldsymbol{T}^{\intercal}\mathbf{1} = \boldsymbol{w}_t,$$

(15)

where $\boldsymbol{D} \in \mathbb{R}_+^{N_s \times N_t}$ is the cost matrix (distance matrix) defining the cost to move mass from source distribution to target distribution, $\boldsymbol{w}_s \in \mathbb{R}^{N_s}$ and $\boldsymbol{w}_t \in \mathbb{R}^{N_t}$ are the weights of each samples in the source and target distribution. The total weights of both $\boldsymbol{w}_s$ and $\boldsymbol{w}_t$ are equal to 1. The objective of the OT problem is to find a transportation plan $\boldsymbol{T}^*$ that minimizes the total transportation cost under the weights-equal constraint.

OT problem has two main functions: (1) Measuring the distance between two distributions; (2) Finding the correspondences between two distributions. We employed both. Specifically, we treated the geographic representations of regions in the source and target cities as two mass distributions, with each region assigned the same weight. we used the solution of the OT problem (also known as the Wasserstein distance) to measure the geographic representation distance of the source and target cities, and treated it as a loss to optimize the spatial encoder, thereby pulling the correspondent regions in representation space.

As illustrated in Equation 15, the OT problem is a linear programming problem, and we use an OT solver based on the network simplex algorithm [2] to address it. Thanks to the Python Optimal Transport (POT) [7] tool, we can conveniently calculate the solution of OT problem.

---

[5] https://www.openstreetmap.org
[6] https://www.worldpop.org/
[7] https://github.com/PythonOT/POT

Table 3: Hyperparameters setting for CRAFT

| Hyperparameter | Setting value |
|---|---|
| Diffusion steps ($n$) | 500 |
| $\beta_1 \sim \beta_n$ | $0.0002 \sim 0.04$ (linear) |
| GraphTransformer (GT) layers | 3 |
| Temporal encoder layers | 2 |
| GT attention heads | 4 |
| Retrieval top-K value | 5 |
| Batch size | 256 |
| Learning rate | $5 \times 10^{-6}$ |
| Training epochs | 300 |
| Regional geographic representation ($\boldsymbol{h}_i$) dimensions | 128 |
| Temporal encoder hidden dimensions (dimensions of $\boldsymbol{x}_{i,t}$) | 256 |
| Hour embedding ($\boldsymbol{t}_{hour}$) dimensions | 64 |
| Weekday embedding ($\boldsymbol{t}_{week}$) dimensions | 64 |
| Month embedding ($\boldsymbol{t}_{month}$) dimensions | 64 |
| Condition ($\boldsymbol{c}_{i,t}$) dimensions | 256 |

## A.3 Implementation Details

For the proposed CRAFT method, we provide the hyperparameter settings in Table 3 to facilitate the reproducibility by researchers. All these parameters are recommended values, not fixed, and can be adjusted according to the dataset and experimental environment. During training, the AdamW optimizer was used. To enhance stability, the EMA (Exponential Moving Average) mechanism was adopted to train the diffusion model.

# B  Details of Experimental Settings

## B.1  Experimental Environment

All neural network models (including CRAFT and other baselines) are implemented in PyTorch and trained on a single NVIDIA RTX 3090 GPU. The experimental machine ran on Ubuntu 20.04.6 LTS, was equipped 24-core Intel(R) Xeon(R) Silver CPU, and had 503 GB of RAM. The training time for all models on a single dataset did not exceed 16 hours.

## B.2  Datasets and Pre-processing

Table 4: Data description

| Datasets | Chicago | Washington D.C. | Toronto | New York City |
|---|---|---|---|---|
| # Trips ($\times 10^3$) | 5136 | 4011 | 2395 | 35080 |
| Time range | 2023.01-2023.12 | 2023.01-2023.12 | 2020.01-2020.12 | 2023.01-2023.12 |
| # Regions | 73 | 82 | 61 | 96 |
| # POIs | 17205 | 14070 | 20621 | 50776 |

We conducted experiments using the traffic flow datasets of four cities, namely Chicago, Washington D.C., Toronto, and New York City. The original data are all trip records of shared bicycles, which include the latitude and longitude of the starting and ending points of users' trips as well as timestamps. We associated the trips with the manually partitioned urban regions, and counted the number of bicycles entering and leaving each region within each hour, which served as the traffic flow values. Details of the datasets are presented in Table 4.

In fact, the user trip data is sparse, which leads to the instability of the traffic flow trend in original data. This is also a common problem in researches about traffic flow data. In response to this, we have adopted two processing methods: (1) We have filled in the missing values in the traffic flow sequence through linear interpolation. For the situation where there are values at the previous and subsequent time steps but missing values in the middle, we have filled them with the average of the

previous and subsequent values. (2) We have used the moving average method to smooth the values within a window with a time length of 3. For the processed data, we have set a sliding window with a length of $T$ to extract the training and test samples. If there are still missing values exceeding $5\%$ in a certain sample, we will discard that sample.

## B.3 Baseline Methods

The baseline models used in the experiment include two types of deep learning traffic flow generation models that have emerged in the literature in recent years: *Static Flow Generation Models* and *Dynamic Flow Generation Models*. *Static Flow Generation Models* directly estimate traffic flow using the geographic features and temporal information of urban regions, and they include the following two models.

- GMEL [27]: It uses two graph neural networks to extract features and can simultaneously predict the inflow and outflow as well as the OD (origin-destination) flow between regions.
- DFG [40]: It uses a deep cross network to extract the POI (point of interest) features inside and outside the region, conducts supervised training using the intention-aware pedestrian flow, and predicts the inflow and outflow of the region.

*Dynamic Flow Generation Models* use deep generative models to learn the complex distribution of data and map random noise into data samples. The selected baseline models are as follows.

- KSTDiff [58]: It uses the urban knowledge graph to extract the representations of urban geographic entities, and then constructs a knowledge-enhanced spatiotemporal diffusion model to generate the inflow and outflow of regions.
- CGAN [29]: This is a conditional generative adversarial network. In the experiment, we use the original static geographic features and time embeddings as conditions to guide the generation of the GAN.
- Diffwave [23]: This is a diffusion model suitable for generating time series data and is often used in speech synthesis. In this experiment, we use it to learn the distribution of traffic flow data.
- DiT [31]: This is a diffusion probabilistic model with a Transformer as the noise estimator.
- DDPM [12]: This is a diffusion probabilistic model U-Net as noise estimator for image synthesizing. To be applied in traffic flow data generation, we replaced 2D convolutions with 1D convolutions.
- CVAE [10]: This is a conditional variational autoencoder. Condition information is added to both the Encoder and the Decoder to guide it to learn the conditional probability distribution of the data. We use the original static geographic features and time embeddings as conditions.

## B.4 Evaluation metrics

We used three commonly used metrics in traffic flow generation research to evaluate the quality of generated data, including Common Part of Commuters (CPC), Normalized Mean Absolute Error (NMAE), Normalized Root Mean Square Error (NRMSE). Since the the actual traffic flow values in different cities vary greatly, the traffic flow data generated by our model are all normalized, which can reflect the relative magnitudes of traffic flows in different regions and at different time in the same city. All the metrics are calculated based on the normalized data, and the calculation formulas are as follows

$$\text{CPC} = \frac{2 \sum_{i=1}^{M} \min{(\hat{y}_i, y_i)}}{\sum_{i=1}^{M} \hat{y}_i + \sum_{i=1}^{M} y_i},$$

$$\text{NMAE} = \frac{\frac{1}{M} \sum_{i=1}^{M} |\hat{y}_i - y_i|}{\max(y_i) - \min(y_i)}, \tag{16}$$

$$\text{NRMSE} = \frac{\sqrt{\frac{1}{M} \sum_{i=1}^{M} (\hat{y}_i - y_i)^2}}{\max(y_i) - \min(y_i)},$$

where $\hat{y}_i$ is the generated value, $y_i$ is the real value, and $M$ is the number of values of all samples in the test dataset. To enhance the stability of evaluation results, we grouped the data by the Region ID, month, weekday and hour of the target city's test samples and calculated the differences between the group-averaged values.

# C  Additional Experiments

## C.1  Sensitivity Analysis Results

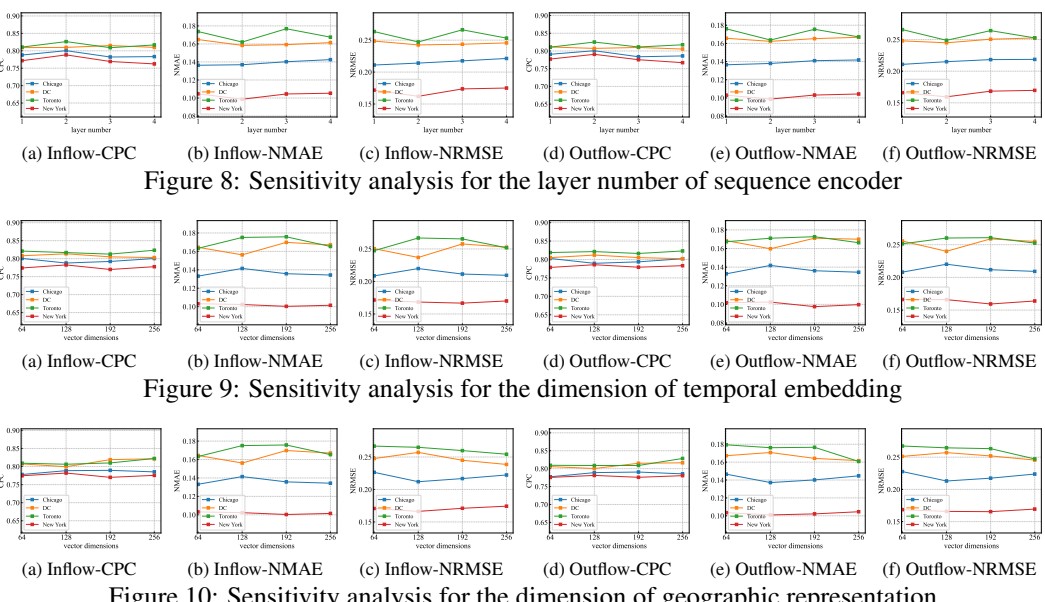

(a) Inflow-CPC    (b) Inflow-NMAE    (c) Inflow-NRMSE    (d) Outflow-CPC    (e) Outflow-NMAE    (f) Outflow-NRMSE

Figure 8: Sensitivity analysis for the layer number of sequence encoder

(a) Inflow-CPC    (b) Inflow-NMAE    (c) Inflow-NRMSE    (d) Outflow-CPC    (e) Outflow-NMAE    (f) Outflow-NRMSE

Figure 9: Sensitivity analysis for the dimension of temporal embedding

(a) Inflow-CPC    (b) Inflow-NMAE    (c) Inflow-NRMSE    (d) Outflow-CPC    (e) Outflow-NMAE    (f) Outflow-NRMSE

Figure 10: Sensitivity analysis for the dimension of geographic representation

To validate how different hyperparameter configurations affect model performance, we plotted the variation curves of evaluation metrics against three key hyperparameters.: (1) the layer number of the sequence encoder; (2) the dimension of temporal embedding; (3) the dimension of the geographic representation. The results of sensitivity analyses across all datasets and evaluation metrics are shown in the Fig. 8, Fig. 9 and Fig. 10. We can observe that CRAFT demonstrates excellent robustness across all datasets and evaluation metrics, eliminating the need for meticulous hyperparameter tuning to ensure superior model performance.

## C.2  Temporal Length Extension Results

**Overall Zero-shot Generation Performance in Target City.** In the main experiments of the paper, we set the generation time window length to $T = 24$. We then progressively extended the time length $T$ to the following values: $\{48, 72, 96, 120, 144, 168\}$. The experimental setup remained consistent: training the model using data from three cities and performing zero-shot generation on the fourth city. Detailed evaluation results are presented in Table 6. From these tables, we can conclude that: (1) CRAFT achieved best performance in over 98.6% of cases across all datasets and evaluation metrics, demonstrating that CRAFT exhibits state-of-the-art (SOTA) zero-shot generalization capabilities for sequence data generation of varying temporal lengths. (2) CRAFT demonstrates stable performance, while methods such as Diffwave and DDPM exhibit significant performance fluctuations under different temporal length settings.

**Data Utility Comparison.** We compared the utility of generated data across different temporal lengths for traffic flow prediction tasks. Regardless of the value of $T$, we used the first $T/2$ historical sequence as input to predict the $T/2$ future flow sequence. The downstream models are trained on generated data from various methods and tested on real data. Detailed results are presented in Table 7 and 8. We found that in 86.5% of cases, the downstream model trained with CRAFT's generated data achieved best performance, indicating that CRAFT's generated data has the better utility than other baselines.

**Computational Cost.** As the length of temporal length increases, the model size and training time will also grow correspondingly. To observe this phenomenon in detail, we conducted experiments on a single NVIDIA RTX 3090 GPU, collected relevant statistical data during the training phase, and the results are shown in Table 5. We observe that our model's computational cost grows linearly with temporal length increases, but this effect is weak. Extending the time length is acceptable in terms of computational cost.

Table 5: The relationship between computational cost and temporal length

| $T$ | Model size (Byte) | Train time (s/epoch) | Valid time (s/epcoh) | Avg memory (GB) |
|-----|-------------------|----------------------|----------------------|-----------------|
| 24  | 51781096          | 40.999               | 5.685                | 0.343           |
| 48  | 51830248          | 41.156               | 5.531                | 0.344           |
| 72  | 51879400          | 43.729               | 6.053                | 0.344           |
| 96  | 51928552          | 44.555               | 5.631                | 0.345           |
| 120 | 51977704          | 46.712               | 6.429                | 0.345           |
| 144 | 52026856          | 48.883               | 6.666                | 0.345           |
| 168 | 52076008          | 51.896               | 6.351                | 0.346           |

## C.3 Visualization of the Traffic Flow Spatial Heatmap

We display the average traffic flow of real data and generated data on maps. The visualizations for all datasets and baselines are shown in Fig. 11. This comparison intuitively demonstrates that the traffic data generated by CRAFT exhibits the highest similarity to real data in terms of spatial distribution, indicating that CRAFT can effectively capture the universal mapping relationship between geographic representations and traffic flow across different cities.

## C.4 Visualization of Geographic Feature Alignment (GFA)

We employed t-SNE analysis to visualize the impact of the Traffic Flow Alignment (TFA) and Cross City Alignment (CCA) modules in Geographic Feature Alignment. The results of the four experiments are presented in Fig. 12. Regardless of which city was chosen as the target, the alignment results exhibited similar conclusions: (1) Under the combined action of TFA and CCA, regions with high and low traffic volumes were well-separated in the representation space. (2) Without TFA, representations from different cities tended to cluster into multiple groups, with high- and low-traffic regions intermingled within the same cluster, making it difficult to distinguish and reducing the quality of conditions. (3) Without CCA, significant domain shift occurred between the source and target cities. Specifically, a portion of the target city's region representations deviated from the concentrated representation area of the source city, potentially leading to poorer transferability.

Table 6: Cross-city traffic flow generation results with extended temporal length

| Method | City | T = 48 CPC | NMAE | NRMSE | T = 72 CPC | NMAE | NRMSE | T = 96 CPC | NMAE | NRMSE | T = 120 CPC | NMAE | NRMSE | T = 144 CPC | NMAE | NRMSE | T = 168 CPC | NMAE | NRMSE |
|---|---|---|---|---|---|---|---|---|---|---|---|---|---|---|---|---|---|---|---|
| GMEL | Chicago(Inflow) | 0.590 | 0.215 | 0.310 | 0.645 | 0.201 | 0.288 | 0.741 | 0.212 | 0.268 | 0.490 | 0.246 | 0.351 | 0.700 | 0.183 | 0.257 | 0.512 | 0.246 | 0.358 |
| DFG | | 0.155 | 0.328 | 0.457 | 0.153 | 0.334 | 0.462 | 0.152 | 0.340 | 0.467 | 0.156 | 0.326 | 0.454 | 0.156 | 0.328 | 0.455 | 0.155 | 0.330 | 0.457 |
| KSTDiff | | 0.000 | 0.358 | 0.488 | 0.460 | 0.288 | 0.399 | 0.003 | 0.370 | 0.499 | 0.540 | 0.444 | 0.533 | 0.552 | 0.539 | 0.616 | 0.032 | 0.360 | 0.489 |
| CGAN | | 0.584 | 0.379 | 0.488 | 0.539 | 0.371 | 0.487 | 0.521 | 0.386 | 0.500 | 0.461 | 0.359 | 0.476 | 0.546 | 0.492 | 0.582 | 0.509 | 0.437 | 0.535 |
| Diffwave | | 0.490 | 0.504 | 0.613 | 0.384 | 0.456 | 0.570 | 0.464 | 0.430 | 0.553 | 0.492 | 0.500 | 0.609 | 0.360 | 0.437 | 0.555 | 0.479 | 0.437 | 0.556 |
| DiT | | 0.572 | 0.325 | 0.407 | 0.564 | 0.362 | 0.448 | 0.593 | 0.333 | 0.405 | 0.580 | 0.356 | 0.445 | 0.561 | 0.341 | 0.425 | 0.510 | 0.340 | 0.438 |
| DDPM | | 0.398 | 0.300 | 0.419 | 0.417 | 0.306 | 0.427 | 0.455 | 0.313 | 0.428 | 0.490 | 0.291 | 0.397 | 0.481 | 0.300 | 0.411 | 0.509 | 0.313 | 0.417 |
| CVAE | | 0.476 | 0.269 | 0.388 | 0.555 | 0.286 | 0.404 | 0.616 | 0.282 | 0.392 | 0.509 | 0.280 | 0.395 | 0.525 | 0.319 | 0.427 | 0.527 | 0.297 | 0.402 |
| **CRAFT** | | **0.789** | **0.145** | **0.223** | **0.798** | **0.145** | **0.224** | **0.791** | **0.150** | **0.229** | **0.785** | **0.149** | **0.228** | **0.777** | **0.154** | **0.233** | **0.645** | **0.251** | **0.356** |
| GMEL | Chicago(Outflow) | 0.667 | 0.197 | 0.278 | 0.455 | 0.266 | 0.373 | 0.720 | 0.199 | 0.259 | 0.606 | 0.213 | 0.310 | 0.429 | 0.266 | 0.371 | 0.502 | 0.248 | 0.352 |
| DFG | | 0.152 | 0.333 | 0.460 | 0.150 | 0.339 | 0.465 | 0.149 | 0.346 | 0.470 | 0.153 | 0.332 | 0.457 | 0.152 | 0.333 | 0.458 | 0.152 | 0.335 | 0.460 |
| KSTDiff | | 0.000 | 0.363 | 0.491 | 0.214 | 0.335 | 0.462 | 0.006 | 0.375 | 0.501 | 0.533 | 0.628 | 0.708 | 0.569 | 0.478 | 0.554 | 0.151 | 0.356 | 0.479 |
| CGAN | | 0.587 | 0.367 | 0.476 | 0.552 | 0.370 | 0.483 | 0.529 | 0.379 | 0.493 | 0.438 | 0.368 | 0.486 | 0.540 | 0.526 | 0.612 | 0.505 | 0.433 | 0.534 |
| Diffwave | | 0.459 | 0.445 | 0.561 | 0.350 | 0.469 | 0.584 | 0.421 | 0.482 | 0.591 | 0.298 | 0.428 | 0.544 | 0.353 | 0.460 | 0.575 | 0.500 | 0.490 | 0.601 |
| DiT | | 0.546 | 0.346 | 0.434 | 0.531 | 0.346 | 0.437 | 0.579 | 0.346 | 0.423 | 0.550 | 0.349 | 0.438 | 0.550 | 0.364 | 0.449 | 0.536 | 0.342 | 0.434 |
| DDPM | | 0.405 | 0.301 | 0.418 | 0.423 | 0.307 | 0.427 | 0.446 | 0.319 | 0.434 | 0.491 | 0.292 | 0.398 | 0.479 | 0.302 | 0.413 | 0.510 | 0.315 | 0.420 |
| CVAE | | 0.472 | 0.274 | 0.392 | 0.562 | 0.287 | 0.404 | 0.620 | 0.283 | 0.392 | 0.507 | 0.281 | 0.395 | 0.534 | 0.316 | 0.422 | 0.532 | 0.294 | 0.398 |
| **CRAFT** | | **0.791** | **0.146** | **0.223** | **0.797** | **0.147** | **0.225** | **0.797** | **0.148** | **0.225** | **0.792** | **0.146** | **0.223** | **0.789** | **0.148** | **0.224** | **0.656** | **0.246** | **0.349** |
| GMEL | Washington, D.C.(Inflow) | 0.728 | 0.234 | 0.284 | 0.609 | 0.289 | 0.383 | 0.649 | 0.272 | 0.360 | 0.764 | 0.223 | 0.261 | 0.726 | 0.227 | 0.292 | 0.711 | 0.234 | 0.301 |
| DFG | | 0.697 | 0.246 | 0.348 | 0.696 | 0.252 | 0.354 | 0.698 | 0.254 | 0.355 | 0.692 | 0.252 | 0.354 | 0.692 | 0.254 | 0.356 | 0.691 | 0.256 | 0.358 |
| KSTDiff | | 0.000 | 0.465 | 0.595 | 0.494 | 0.356 | 0.458 | 0.289 | 0.421 | 0.542 | 0.039 | 0.464 | 0.591 | 0.000 | 0.477 | 0.603 | 0.641 | 0.477 | 0.584 |
| CGAN | | 0.596 | 0.342 | 0.465 | 0.510 | 0.420 | 0.538 | 0.575 | 0.440 | 0.550 | 0.583 | 0.391 | 0.507 | 0.653 | 0.480 | 0.593 | 0.442 | 0.391 | 0.514 |
| Diffwave | | 0.578 | 0.505 | 0.624 | 0.274 | 0.466 | 0.584 | 0.407 | 0.557 | 0.663 | 0.450 | 0.577 | 0.677 | 0.641 | 0.506 | 0.623 | 0.457 | 0.580 | 0.673 |
| DiT | | 0.621 | 0.351 | 0.440 | 0.627 | 0.354 | 0.432 | 0.617 | 0.369 | 0.446 | 0.608 | 0.359 | 0.451 | 0.598 | 0.386 | 0.479 | 0.630 | 0.357 | 0.437 |
| DDPM | | 0.476 | 0.355 | 0.477 | 0.417 | 0.418 | 0.534 | 0.488 | 0.403 | 0.510 | 0.541 | 0.349 | 0.458 | 0.609 | 0.340 | 0.443 | 0.463 | 0.370 | 0.475 |
| CVAE | | 0.296 | 0.402 | 0.534 | 0.480 | 0.368 | 0.486 | 0.423 | 0.388 | 0.511 | 0.566 | 0.347 | 0.455 | 0.433 | 0.399 | 0.518 | 0.464 | 0.374 | 0.491 |
| **CRAFT** | | **0.791** | **0.185** | **0.275** | **0.786** | **0.191** | **0.282** | **0.777** | **0.202** | **0.301** | **0.781** | **0.191** | **0.281** | **0.781** | **0.194** | **0.285** | **0.778** | **0.197** | **0.288** |
| GMEL | Washington, D.C.(Outflow) | 0.683 | 0.239 | 0.321 | 0.557 | 0.303 | 0.401 | 0.616 | 0.297 | 0.390 | 0.534 | 0.310 | 0.408 | 0.633 | 0.271 | 0.363 | 0.688 | 0.245 | 0.317 |
| DFG | | 0.696 | 0.246 | 0.347 | 0.694 | 0.252 | 0.353 | 0.695 | 0.254 | 0.355 | 0.689 | 0.252 | 0.353 | 0.689 | 0.254 | 0.355 | 0.688 | 0.256 | 0.357 |
| KSTDiff | | 0.000 | 0.470 | 0.598 | 0.581 | 0.338 | 0.412 | 0.004 | 0.488 | 0.612 | 0.000 | 0.476 | 0.601 | 0.536 | 0.503 | 0.622 | 0.653 | 0.513 | 0.629 |
| CGAN | | 0.583 | 0.354 | 0.476 | 0.543 | 0.427 | 0.540 | 0.596 | 0.441 | 0.551 | 0.554 | 0.408 | 0.519 | 0.653 | 0.497 | 0.610 | 0.423 | 0.401 | 0.519 |
| Diffwave | | 0.502 | 0.481 | 0.605 | 0.321 | 0.434 | 0.562 | 0.254 | 0.546 | 0.650 | 0.289 | 0.476 | 0.596 | 0.202 | 0.546 | 0.655 | 0.450 | 0.421 | 0.549 |
| DiT | | 0.580 | 0.371 | 0.461 | 0.579 | 0.361 | 0.450 | 0.623 | 0.377 | 0.464 | 0.583 | 0.357 | 0.452 | 0.589 | 0.387 | 0.478 | 0.621 | 0.372 | 0.463 |
| DDPM | | 0.474 | 0.359 | 0.479 | 0.422 | 0.416 | 0.531 | 0.496 | 0.395 | 0.503 | 0.543 | 0.353 | 0.460 | 0.607 | 0.345 | 0.448 | 0.462 | 0.372 | 0.475 |
| CVAE | | 0.301 | 0.404 | 0.533 | 0.484 | 0.368 | 0.483 | 0.429 | 0.386 | 0.507 | 0.551 | 0.355 | 0.461 | 0.436 | 0.400 | 0.516 | 0.464 | 0.376 | 0.489 |
| **CRAFT** | | **0.790** | **0.188** | **0.277** | **0.783** | **0.195** | **0.285** | **0.773** | **0.207** | **0.304** | **0.781** | **0.193** | **0.280** | **0.781** | **0.196** | **0.284** | **0.778** | **0.199** | **0.288** |
| GMEL | Toronto(Inflow) | 0.748 | 0.230 | 0.281 | 0.721 | 0.240 | 0.309 | 0.639 | 0.283 | 0.366 | 0.558 | 0.313 | 0.407 | 0.672 | 0.260 | 0.344 | 0.703 | 0.255 | 0.326 |
| DFG | | 0.275 | 0.421 | 0.535 | 0.271 | 0.432 | 0.544 | 0.269 | 0.441 | 0.551 | 0.275 | 0.423 | 0.536 | 0.274 | 0.425 | 0.537 | 0.272 | 0.430 | 0.542 |
| KSTDiff | | 0.002 | 0.500 | 0.622 | 0.672 | 0.398 | 0.472 | 0.685 | 0.479 | 0.606 | 0.670 | 0.393 | 0.468 | 0.378 | 0.427 | 0.536 | 0.268 | 0.485 | 0.601 |
| CGAN | | 0.631 | 0.358 | 0.474 | 0.394 | 0.436 | 0.546 | 0.528 | 0.399 | 0.513 | 0.546 | 0.378 | 0.478 | 0.556 | 0.387 | 0.495 | 0.421 | 0.404 | 0.512 |
| Diffwave | | 0.550 | 0.424 | 0.553 | 0.634 | 0.462 | 0.588 | 0.498 | 0.523 | 0.633 | 0.567 | 0.475 | 0.594 | 0.506 | 0.488 | 0.605 | 0.583 | 0.471 | 0.587 |
| DiT | | 0.638 | 0.381 | 0.469 | 0.608 | 0.378 | 0.458 | 0.595 | 0.388 | 0.477 | 0.600 | 0.377 | 0.460 | 0.597 | 0.378 | 0.467 | 0.611 | 0.377 | 0.460 |
| DDPM | | 0.625 | 0.360 | 0.471 | 0.649 | 0.367 | 0.479 | 0.600 | 0.388 | 0.498 | 0.628 | 0.375 | 0.484 | 0.608 | 0.373 | 0.483 | 0.602 | 0.381 | 0.490 |
| CVAE | | 0.714 | 0.302 | 0.402 | 0.712 | 0.305 | 0.414 | 0.699 | 0.305 | 0.412 | 0.679 | 0.302 | 0.416 | 0.583 | 0.368 | 0.469 | 0.592 | 0.368 | 0.461 |
| **CRAFT** | | **0.817** | **0.178** | **0.265** | **0.824** | **0.176** | **0.264** | **0.826** | **0.174** | **0.260** | **0.811** | **0.181** | **0.269** | **0.816** | **0.182** | **0.273** | **0.800** | **0.194** | **0.289** |
| GMEL | Toronto(Outflow) | 0.736 | 0.235 | 0.292 | 0.632 | 0.286 | 0.370 | 0.716 | 0.248 | 0.317 | 0.734 | 0.236 | 0.297 | 0.688 | 0.254 | 0.334 | 0.722 | 0.240 | 0.308 |
| DFG | | 0.274 | 0.420 | 0.533 | 0.270 | 0.432 | 0.543 | 0.268 | 0.441 | 0.550 | 0.274 | 0.422 | 0.534 | 0.274 | 0.425 | 0.536 | 0.271 | 0.430 | 0.540 |
| KSTDiff | | 0.002 | 0.498 | 0.619 | 0.682 | 0.411 | 0.492 | 0.685 | 0.479 | 0.604 | 0.673 | 0.402 | 0.482 | 0.646 | 0.419 | 0.513 | 0.310 | 0.492 | 0.609 |
| CGAN | | 0.637 | 0.359 | 0.472 | 0.523 | 0.420 | 0.526 | 0.526 | 0.405 | 0.515 | 0.544 | 0.379 | 0.478 | 0.547 | 0.393 | 0.499 | 0.559 | 0.368 | 0.457 |
| Diffwave | | 0.469 | 0.432 | 0.548 | 0.617 | 0.446 | 0.564 | 0.558 | 0.438 | 0.561 | 0.420 | 0.471 | 0.589 | 0.574 | 0.432 | 0.547 | 0.555 | 0.467 | 0.581 |
| DiT | | 0.619 | 0.381 | 0.469 | 0.588 | 0.394 | 0.482 | 0.619 | 0.381 | 0.465 | 0.611 | 0.365 | 0.444 | 0.595 | 0.397 | 0.489 | 0.604 | 0.380 | 0.467 |
| DDPM | | 0.628 | 0.360 | 0.470 | 0.650 | 0.373 | 0.485 | 0.603 | 0.390 | 0.498 | 0.633 | 0.375 | 0.483 | 0.609 | 0.376 | 0.486 | 0.607 | 0.382 | 0.490 |
| CVAE | | 0.709 | 0.307 | 0.408 | 0.705 | 0.313 | 0.420 | 0.695 | 0.310 | 0.415 | 0.680 | 0.305 | 0.417 | 0.591 | 0.365 | 0.463 | 0.606 | 0.358 | 0.449 |
| **CRAFT** | | **0.814** | **0.182** | **0.270** | **0.821** | **0.180** | **0.270** | **0.822** | **0.181** | **0.269** | **0.810** | **0.182** | **0.271** | **0.814** | **0.188** | **0.280** | **0.803** | **0.195** | **0.289** |
| GMEL | New York City(Inflow) | 0.676 | 0.178 | 0.238 | 0.621 | 0.256 | 0.336 | 0.598 | 0.297 | 0.362 | 0.611 | 0.263 | 0.331 | 0.602 | 0.282 | 0.347 | 0.511 | 0.429 | 0.500 |
| DFG | | 0.584 | 0.182 | 0.279 | 0.585 | 0.182 | 0.279 | 0.584 | 0.182 | 0.279 | 0.580 | 0.181 | 0.278 | 0.582 | 0.181 | 0.278 | 0.583 | 0.181 | 0.277 |
| KSTDiff | | 0.355 | 0.442 | 0.566 | 0.035 | 0.232 | 0.368 | 0.000 | 0.234 | 0.370 | 0.372 | 0.771 | 0.822 | 0.169 | 0.226 | 0.356 | 0.184 | 0.236 | 0.361 |
| CGAN | | 0.425 | 0.561 | 0.667 | 0.419 | 0.585 | 0.682 | 0.396 | 0.610 | 0.703 | 0.458 | 0.437 | 0.559 | 0.411 | 0.549 | 0.654 | 0.444 | 0.380 | 0.496 |
| Diffwave | | 0.313 | 0.500 | 0.621 | 0.213 | 0.388 | 0.531 | 0.376 | 0.648 | 0.735 | 0.413 | 0.405 | 0.548 | 0.236 | 0.435 | 0.576 | 0.219 | 0.540 | 0.661 |
| DiT | | 0.470 | 0.384 | 0.459 | 0.441 | 0.324 | 0.422 | 0.470 | 0.335 | 0.420 | 0.440 | 0.398 | 0.480 | 0.471 | 0.351 | 0.435 | 0.447 | 0.375 | 0.464 |
| DDPM | | 0.517 | 0.375 | 0.499 | 0.481 | 0.281 | 0.389 | 0.519 | 0.247 | 0.362 | 0.485 | 0.335 | 0.454 | 0.484 | 0.330 | 0.445 | 0.456 | 0.306 | 0.421 |
| CVAE | | 0.442 | 0.406 | 0.535 | 0.483 | 0.324 | 0.454 | 0.472 | 0.343 | 0.471 | 0.286 | 0.273 | 0.406 | 0.485 | 0.286 | 0.406 | 0.470 | 0.329 | 0.437 |
| **CRAFT** | | **0.769** | **0.115** | **0.184** | **0.758** | **0.121** | **0.193** | **0.757** | **0.122** | **0.192** | **0.761** | **0.118** | **0.187** | **0.737** | **0.135** | **0.208** | **0.741** | **0.131** | **0.203** |
| GMEL | New York City(Outflow) | 0.636 | 0.221 | 0.288 | 0.623 | 0.242 | 0.311 | 0.412 | 0.584 | 0.647 | 0.496 | 0.453 | 0.537 | 0.495 | 0.397 | 0.477 | 0.521 | 0.373 | 0.453 |
| DFG | | 0.589 | 0.183 | 0.279 | 0.589 | 0.183 | 0.279 | 0.589 | 0.183 | 0.279 | 0.585 | 0.182 | 0.277 | 0.587 | 0.182 | 0.278 | 0.587 | 0.182 | 0.278 |
| KSTDiff | | 0.381 | 0.763 | 0.815 | 0.157 | 0.229 | 0.359 | 0.000 | 0.234 | 0.370 | 0.428 | 0.429 | 0.522 | 0.349 | 0.238 | 0.345 | 0.282 | 0.264 | 0.387 |
| CGAN | | 0.422 | 0.579 | 0.681 | 0.414 | 0.591 | 0.685 | 0.394 | 0.620 | 0.710 | 0.460 | 0.444 | 0.562 | 0.408 | 0.532 | 0.640 | 0.439 | 0.355 | 0.478 |
| Diffwave | | 0.412 | 0.530 | 0.648 | 0.285 | 0.445 | 0.579 | 0.420 | 0.488 | 0.616 | 0.463 | 0.419 | 0.557 | 0.349 | 0.457 | 0.587 | 0.192 | 0.345 | 0.489 |
| DiT | | 0.436 | 0.359 | 0.444 | 0.460 | 0.346 | 0.426 | 0.485 | 0.353 | 0.425 | 0.418 | 0.362 | 0.459 | 0.477 | 0.385 | 0.471 | 0.457 | 0.342 | 0.422 |
| DDPM | | 0.521 | 0.375 | 0.499 | 0.494 | 0.291 | 0.398 | 0.521 | 0.248 | 0.359 | 0.493 | 0.341 | 0.460 | 0.492 | 0.333 | 0.448 | 0.461 | 0.308 | 0.422 |
| CVAE | | 0.449 | 0.421 | 0.547 | 0.477 | 0.324 | 0.455 | 0.479 | 0.358 | 0.483 | 0.294 | 0.271 | 0.402 | 0.492 | 0.285 | 0.404 | 0.475 | 0.320 | 0.424 |
| **CRAFT** | | **0.777** | **0.112** | **0.176** | **0.766** | **0.119** | **0.185** | **0.764** | **0.121** | **0.186** | **0.762** | **0.118** | **0.184** | **0.743** | **0.136** | **0.203** | **0.746** | **0.134** | **0.200** |

Table 7: Data utility comparison on traffic flow prediction (LSTM)

| Gen | Pred | Chicago | | | | Washington, D.C. | | | | Toronto | | | | New York City | | | |
|---|---|---|---|---|---|---|---|---|---|---|---|---|---|---|---|---|---|
| | | Inflow | | Outflow | | Inflow | | Outflow | | Inflow | | Outflow | | Inflow | | Outflow | |
| | | MAE | RMSE | MAE | RMSE | MAE | RMSE | MAE | RMSE | MAE | RMSE | MAE | RMSE | MAE | RMSE | MAE | RMSE |
| Real | LSTM (T = 48) | 0.097 | 0.151 | 0.100 | 0.154 | 0.094 | 0.144 | 0.097 | 0.150 | 0.109 | 0.163 | 0.111 | 0.164 | 0.056 | 0.101 | 0.056 | 0.101 |
| GMEL | | 0.167 | 0.230 | 0.166 | 0.228 | 0.202 | 0.254 | 0.210 | 0.268 | 0.241 | 0.303 | 0.251 | 0.320 | 0.092 | 0.144 | 0.091 | 0.142 |
| DFG | | 0.283 | 0.418 | 0.289 | 0.423 | 0.134 | 0.202 | 0.136 | 0.206 | 0.319 | 0.426 | 0.321 | 0.427 | 0.070 | 0.126 | 0.069 | 0.125 |
| KSTDiff | | 0.362 | 0.493 | 0.368 | 0.497 | 0.469 | 0.599 | 0.475 | 0.603 | 0.507 | 0.629 | 0.505 | 0.627 | 0.348 | 0.389 | 0.762 | 0.814 |
| CGAN | | 0.199 | 0.292 | 0.212 | 0.309 | 0.283 | 0.394 | 0.286 | 0.399 | 0.287 | 0.388 | 0.286 | 0.382 | 0.151 | 0.248 | 0.163 | 0.257 |
| Diffwave | | 0.221 | 0.299 | 0.232 | 0.297 | 0.276 | 0.354 | 0.331 | 0.441 | 0.327 | 0.434 | 0.286 | 0.362 | 0.141 | 0.211 | 0.138 | 0.211 |
| DiT | | 0.232 | 0.285 | 0.246 | 0.292 | 0.266 | 0.311 | 0.288 | 0.330 | 0.299 | 0.337 | 0.307 | 0.343 | 0.252 | 0.280 | 0.209 | 0.245 |
| DDPM | | 0.113 | 0.174 | 0.115 | 0.171 | 0.124 | 0.183 | 0.122 | 0.183 | **0.117** | **0.171** | **0.122** | **0.175** | 0.065 | 0.113 | 0.066 | 0.110 |
| CVAE | | 0.175 | 0.255 | 0.172 | 0.249 | 0.308 | 0.429 | 0.304 | 0.421 | 0.158 | 0.230 | 0.162 | 0.233 | 0.138 | 0.219 | 0.135 | 0.213 |
| **CRAFT** | | **0.099** | **0.153** | **0.103** | **0.158** | **0.103** | **0.156** | **0.111** | **0.167** | 0.120 | 0.173 | 0.127 | 0.180 | **0.063** | **0.107** | **0.064** | **0.105** |
| Real | LSTM (T = 72) | 0.104 | 0.156 | 0.107 | 0.159 | 0.095 | 0.145 | 0.099 | 0.151 | 0.114 | 0.169 | 0.115 | 0.170 | 0.059 | 0.106 | 0.059 | 0.105 |
| GMEL | | 0.169 | 0.234 | 0.175 | 0.246 | 0.227 | 0.285 | 0.240 | 0.302 | 0.258 | 0.329 | 0.270 | 0.349 | 0.093 | 0.140 | 0.097 | 0.153 |
| DFG | | 0.303 | 0.435 | 0.306 | 0.438 | 0.142 | 0.213 | 0.151 | 0.226 | 0.368 | 0.481 | 0.370 | 0.482 | 0.073 | 0.129 | 0.073 | 0.129 |
| KSTDiff | | 0.256 | 0.340 | 0.282 | 0.381 | 0.320 | 0.400 | 0.314 | 0.372 | 0.336 | 0.380 | 0.340 | 0.403 | 0.224 | 0.361 | 0.205 | 0.343 |
| CGAN | | 0.208 | 0.300 | 0.206 | 0.294 | 0.294 | 0.401 | 0.294 | 0.396 | 0.407 | 0.510 | 0.409 | 0.519 | 0.262 | 0.327 | 0.267 | 0.335 |
| Diffwave | | 0.244 | 0.310 | 0.239 | 0.342 | 0.312 | 0.352 | 0.296 | 0.363 | 0.360 | 0.462 | 0.405 | 0.492 | 0.175 | 0.236 | 0.142 | 0.214 |
| DiT | | 0.259 | 0.300 | 0.228 | 0.283 | 0.292 | 0.332 | 0.306 | 0.354 | 0.321 | 0.366 | 0.312 | 0.355 | 0.169 | 0.231 | 0.204 | 0.244 |
| DDPM | | 0.116 | 0.171 | 0.117 | 0.176 | 0.133 | 0.186 | 0.134 | 0.190 | **0.121** | **0.177** | 0.127 | 0.183 | 0.072 | 0.122 | 0.074 | 0.119 |
| CVAE | | 0.175 | 0.256 | 0.177 | 0.258 | 0.229 | 0.319 | 0.233 | 0.321 | 0.174 | 0.248 | 0.177 | 0.250 | 0.108 | 0.178 | 0.109 | 0.180 |
| **CRAFT** | | **0.109** | **0.165** | **0.111** | **0.167** | **0.111** | **0.166** | **0.116** | **0.171** | 0.124 | 0.179 | 0.127 | 0.180 | **0.066** | **0.111** | **0.067** | **0.108** |
| Real | LSTM (T = 96) | 0.108 | 0.161 | 0.110 | 0.163 | 0.094 | 0.144 | 0.098 | 0.149 | 0.117 | 0.175 | 0.119 | 0.175 | 0.062 | 0.111 | 0.063 | 0.110 |
| GMEL | | 0.174 | 0.245 | 0.182 | 0.248 | 0.231 | 0.288 | 0.248 | 0.312 | 0.261 | 0.319 | 0.262 | 0.313 | 0.102 | 0.152 | 0.139 | 0.198 |
| DFG | | 0.322 | 0.460 | 0.326 | 0.462 | 0.152 | 0.224 | 0.158 | 0.230 | 0.351 | 0.455 | 0.351 | 0.454 | 0.081 | 0.140 | 0.081 | 0.142 |
| KSTDiff | | 0.371 | 0.502 | 0.376 | 0.505 | 0.416 | 0.540 | 0.494 | 0.617 | 0.480 | 0.608 | 0.480 | 0.606 | 0.234 | 0.372 | 0.235 | 0.371 |
| CGAN | | 0.261 | 0.372 | 0.264 | 0.376 | 0.397 | 0.515 | 0.409 | 0.519 | 0.335 | 0.453 | 0.336 | 0.454 | 0.170 | 0.240 | 0.164 | 0.232 |
| Diffwave | | 0.252 | 0.341 | 0.236 | 0.321 | 0.319 | 0.378 | 0.297 | 0.350 | 0.311 | 0.384 | 0.327 | 0.412 | 0.149 | 0.229 | 0.182 | 0.271 |
| DiT | | 0.234 | 0.287 | 0.242 | 0.287 | 0.297 | 0.335 | 0.296 | 0.333 | 0.316 | 0.357 | 0.302 | 0.341 | 0.230 | 0.260 | 0.230 | 0.260 |
| DDPM | | 0.116 | 0.172 | 0.119 | 0.177 | 0.122 | 0.179 | 0.122 | 0.184 | **0.128** | **0.183** | 0.135 | 0.190 | 0.074 | 0.125 | 0.074 | 0.118 |
| CVAE | | 0.152 | 0.225 | 0.159 | 0.233 | 0.301 | 0.405 | 0.300 | 0.402 | 0.170 | 0.240 | 0.174 | 0.241 | 0.129 | 0.202 | 0.131 | 0.203 |
| **CRAFT** | | **0.112** | **0.170** | **0.115** | **0.171** | **0.111** | **0.162** | **0.115** | **0.171** | 0.129 | 0.188 | 0.134 | 0.191 | **0.069** | **0.118** | **0.068** | **0.112** |
| Real | LSTM (T = 120) | 0.106 | 0.160 | 0.108 | 0.161 | 0.095 | 0.145 | 0.099 | 0.149 | 0.121 | 0.183 | 0.122 | 0.183 | 0.062 | 0.111 | 0.063 | 0.110 |
| GMEL | | 0.184 | 0.243 | 0.187 | 0.256 | 0.213 | 0.260 | 0.218 | 0.271 | 0.277 | 0.333 | 0.262 | 0.312 | 0.099 | 0.152 | 0.105 | 0.155 |
| DFG | | 0.306 | 0.445 | 0.311 | 0.448 | 0.168 | 0.249 | 0.172 | 0.252 | 0.352 | 0.467 | 0.356 | 0.472 | 0.089 | 0.154 | 0.089 | 0.154 |
| KSTDiff | | 0.390 | 0.434 | 0.624 | 0.702 | 0.458 | 0.588 | 0.473 | 0.599 | 0.334 | 0.377 | 0.337 | 0.388 | 0.767 | 0.819 | 0.390 | 0.444 |
| CGAN | | 0.275 | 0.374 | 0.283 | 0.387 | 0.293 | 0.404 | 0.286 | 0.395 | 0.314 | 0.409 | 0.313 | 0.410 | 0.136 | 0.228 | 0.137 | 0.227 |
| Diffwave | | 0.236 | 0.294 | 0.233 | 0.299 | 0.387 | 0.495 | 0.324 | 0.413 | 0.307 | 0.383 | 0.294 | 0.358 | 0.151 | 0.245 | 0.143 | 0.235 |
| DiT | | 0.231 | 0.284 | 0.242 | 0.286 | 0.276 | 0.319 | 0.277 | 0.321 | 0.302 | 0.344 | 0.299 | 0.344 | 0.270 | 0.297 | 0.175 | 0.219 |
| DDPM | | 0.116 | 0.172 | 0.118 | 0.177 | 0.123 | 0.181 | 0.122 | 0.181 | 0.132 | 0.188 | 0.141 | 0.196 | 0.070 | 0.118 | 0.069 | 0.112 |
| CVAE | | 0.151 | 0.217 | 0.155 | 0.222 | 0.214 | 0.304 | 0.220 | 0.310 | 0.166 | 0.239 | 0.164 | 0.234 | 0.133 | 0.215 | 0.130 | 0.208 |
| **CRAFT** | | **0.110** | **0.168** | **0.112** | **0.170** | **0.110** | **0.166** | **0.116** | **0.170** | 0.129 | 0.187 | 0.131 | 0.190 | 0.073 | 0.115 | 0.074 | 0.114 |
| Real | LSTM (T = 144) | 0.107 | 0.160 | 0.110 | 0.162 | 0.095 | 0.144 | 0.098 | 0.148 | 0.125 | 0.186 | 0.127 | 0.186 | 0.062 | 0.109 | 0.061 | 0.108 |
| GMEL | | 0.184 | 0.242 | 0.221 | 0.327 | 0.209 | 0.262 | 0.238 | 0.303 | 0.261 | 0.317 | 0.263 | 0.317 | 0.086 | 0.137 | 0.091 | 0.138 |
| DFG | | 0.315 | 0.454 | 0.319 | 0.456 | 0.155 | 0.227 | 0.157 | 0.229 | 0.368 | 0.487 | 0.372 | 0.490 | 0.086 | 0.145 | 0.086 | 0.145 |
| KSTDiff | | 0.524 | 0.585 | 0.456 | 0.505 | 0.481 | 0.607 | 0.350 | 0.386 | 0.341 | 0.419 | 0.377 | 0.423 | 0.200 | 0.334 | 0.188 | 0.294 |
| CGAN | | 0.394 | 0.495 | 0.420 | 0.515 | 0.459 | 0.579 | 0.470 | 0.590 | 0.362 | 0.460 | 0.355 | 0.452 | 0.172 | 0.255 | 0.176 | 0.260 |
| Diffwave | | 0.244 | 0.335 | 0.312 | 0.425 | 0.290 | 0.358 | 0.295 | 0.346 | 0.333 | 0.440 | 0.366 | 0.459 | 0.167 | 0.251 | 0.142 | 0.210 |
| DiT | | 0.226 | 0.280 | 0.224 | 0.276 | 0.306 | 0.345 | 0.305 | 0.342 | 0.304 | 0.344 | 0.311 | 0.352 | 0.195 | 0.234 | 0.206 | 0.240 |
| DDPM | | 0.117 | 0.176 | 0.121 | 0.183 | 0.123 | 0.182 | 0.125 | 0.185 | 0.132 | **0.188** | 0.137 | **0.194** | 0.076 | 0.121 | 0.078 | 0.117 |
| CVAE | | 0.209 | 0.293 | 0.210 | 0.293 | 0.278 | 0.370 | 0.279 | 0.370 | 0.226 | 0.314 | 0.228 | 0.316 | 0.154 | 0.255 | 0.153 | 0.252 |
| **CRAFT** | | **0.113** | **0.170** | **0.115** | **0.170** | **0.114** | **0.169** | **0.118** | **0.171** | **0.130** | 0.189 | 0.137 | 0.196 | 0.075 | 0.122 | 0.077 | 0.118 |
| Real | LSTM (T = 168) | 0.112 | 0.169 | 0.115 | 0.172 | 0.098 | 0.146 | 0.101 | 0.152 | 0.121 | 0.179 | 0.122 | 0.179 | 0.061 | 0.107 | 0.061 | 0.107 |
| GMEL | | 0.204 | 0.280 | 0.205 | 0.283 | 0.198 | 0.256 | 0.211 | 0.258 | 0.225 | 0.268 | 0.226 | 0.279 | 0.098 | 0.148 | 0.097 | 0.148 |
| DFG | | 0.299 | 0.431 | 0.304 | 0.433 | 0.153 | 0.223 | 0.156 | 0.228 | 0.312 | 0.414 | 0.313 | 0.414 | 0.089 | 0.154 | 0.089 | 0.153 |
| KSTDiff | | 0.348 | 0.480 | 0.335 | 0.462 | 0.443 | 0.524 | 0.505 | 0.623 | 0.401 | 0.516 | 0.379 | 0.481 | 0.199 | 0.339 | 0.195 | 0.305 |
| CGAN | | 0.412 | 0.504 | 0.426 | 0.525 | 0.395 | 0.496 | 0.390 | 0.484 | 0.335 | 0.412 | 0.340 | 0.416 | 0.181 | 0.269 | 0.173 | 0.261 |
| Diffwave | | 0.238 | 0.306 | 0.227 | 0.295 | 0.334 | 0.388 | 0.444 | 0.573 | 0.307 | 0.354 | 0.331 | 0.407 | 0.166 | 0.260 | 0.147 | 0.220 |
| DiT | | 0.235 | 0.296 | 0.236 | 0.295 | 0.289 | 0.326 | 0.285 | 0.324 | 0.315 | 0.351 | 0.317 | 0.357 | 0.218 | 0.250 | 0.168 | 0.214 |
| DDPM | | 0.118 | 0.175 | 0.121 | 0.182 | 0.131 | 0.181 | 0.134 | 0.188 | 0.125 | 0.182 | 0.137 | 0.195 | **0.073** | **0.119** | **0.075** | **0.119** |
| CVAE | | 0.228 | 0.312 | 0.228 | 0.311 | 0.316 | 0.416 | 0.313 | 0.411 | 0.187 | 0.251 | 0.190 | 0.251 | 0.180 | 0.260 | 0.180 | 0.256 |
| **CRAFT** | | **0.114** | **0.170** | **0.117** | **0.173** | **0.111** | **0.165** | **0.115** | **0.171** | **0.124** | **0.180** | **0.133** | **0.188** | 0.078 | 0.128 | 0.078 | 0.122 |

Table 8: Data utility comparison on traffic flow prediction (Transformer)

| Gen | Pred | Chicago Inflow MAE | RMSE | Outflow MAE | RMSE | Washington, D.C. Inflow MAE | RMSE | Outflow MAE | RMSE | Toronto Inflow MAE | RMSE | Outflow MAE | RMSE | New York City Inflow MAE | RMSE | Outflow MAE | RMSE |
|---|---|---|---|---|---|---|---|---|---|---|---|---|---|---|---|---|---|
| Real | Transformer (T = 48) | 0.099 | 0.158 | 0.103 | 0.162 | 0.094 | 0.149 | 0.098 | 0.155 | 0.111 | 0.170 | 0.114 | 0.172 | 0.053 | 0.102 | 0.054 | 0.103 |
| GMEL | | 0.165 | 0.228 | 0.165 | 0.227 | 0.201 | 0.260 | 0.199 | 0.253 | 0.220 | 0.274 | 0.223 | 0.284 | 0.080 | 0.133 | 0.081 | 0.131 |
| DFG | | 0.300 | 0.424 | 0.300 | 0.418 | 0.184 | 0.250 | 0.176 | 0.240 | 0.341 | 0.453 | 0.328 | 0.435 | 0.085 | 0.134 | 0.081 | 0.130 |
| KSTDiff | | 0.367 | 0.500 | 0.377 | 0.510 | 0.469 | 0.604 | 0.476 | 0.609 | 0.571 | 0.723 | 0.530 | 0.670 | 0.337 | 0.378 | 0.756 | 0.807 |
| CGAN | | 0.159 | 0.221 | 0.171 | 0.237 | 0.282 | 0.379 | 0.284 | 0.382 | 0.239 | 0.310 | 0.235 | 0.304 | 0.128 | 0.193 | 0.135 | 0.201 |
| Diffwave | | 0.220 | 0.292 | 0.212 | 0.279 | 0.278 | 0.345 | 0.284 | 0.354 | 0.291 | 0.356 | 0.285 | 0.347 | 0.156 | 0.228 | 0.140 | 0.211 |
| DiT | | 0.235 | 0.289 | 0.236 | 0.287 | 0.269 | 0.312 | 0.282 | 0.320 | 0.296 | 0.339 | 0.298 | 0.337 | 0.181 | 0.225 | 0.158 | 0.212 |
| DDPM | | 0.111 | 0.169 | 0.115 | 0.173 | 0.113 | 0.166 | 0.115 | 0.172 | **0.119** | **0.174** | **0.123** | 0.178 | 0.063 | 0.111 | 0.065 | 0.110 |
| CVAE | | 0.162 | 0.228 | 0.164 | 0.229 | 0.230 | 0.311 | 0.237 | 0.318 | 0.172 | 0.222 | 0.176 | 0.226 | 0.114 | 0.181 | 0.116 | 0.183 |
| **CRAFT** | | **0.097** | **0.148** | **0.100** | **0.152** | **0.097** | **0.149** | **0.105** | **0.161** | 0.124 | 0.174 | 0.124 | 0.173 | **0.060** | **0.103** | **0.060** | **0.102** |
| Real | Transformer (T = 72) | 0.102 | 0.159 | 0.105 | 0.162 | 0.094 | 0.147 | 0.098 | 0.153 | 0.116 | 0.176 | 0.118 | 0.177 | 0.055 | 0.100 | 0.056 | 0.100 |
| GMEL | | 0.186 | 0.251 | 0.186 | 0.258 | 0.249 | 0.311 | 0.240 | 0.298 | 0.252 | 0.303 | 0.248 | 0.298 | 0.083 | 0.138 | 0.083 | 0.132 |
| DFG | | 0.362 | 0.491 | 0.350 | 0.486 | 0.166 | 0.222 | 0.166 | 0.222 | 0.391 | 0.511 | 0.390 | 0.509 | 0.069 | 0.116 | 0.070 | 0.118 |
| KSTDiff | | 0.349 | 0.514 | 0.364 | 0.519 | 0.383 | 0.511 | 0.376 | 0.488 | 0.356 | 0.401 | 0.352 | 0.413 | 0.216 | 0.355 | 0.207 | 0.359 |
| CGAN | | 0.179 | 0.243 | 0.187 | 0.249 | 0.254 | 0.345 | 0.252 | 0.333 | 0.391 | 0.493 | 0.359 | 0.458 | 0.146 | 0.190 | 0.149 | 0.194 |
| Diffwave | | 0.239 | 0.297 | 0.228 | 0.294 | 0.299 | 0.352 | 0.279 | 0.328 | 0.324 | 0.369 | 0.339 | 0.382 | 0.143 | 0.211 | 0.143 | 0.209 |
| DiT | | 0.243 | 0.291 | 0.225 | 0.281 | 0.285 | 0.328 | 0.287 | 0.331 | 0.321 | 0.364 | 0.311 | 0.354 | 0.177 | 0.234 | 0.188 | 0.234 |
| DDPM | | 0.108 | 0.161 | 0.113 | 0.167 | 0.132 | 0.187 | 0.131 | 0.186 | 0.128 | 0.184 | 0.130 | 0.185 | 0.067 | 0.117 | 0.070 | 0.118 |
| CVAE | | 0.175 | 0.243 | 0.176 | 0.244 | 0.224 | 0.305 | 0.227 | 0.306 | 0.192 | 0.255 | 0.198 | 0.259 | 0.106 | 0.165 | 0.109 | 0.170 |
| **CRAFT** | | **0.103** | **0.155** | **0.106** | **0.158** | **0.106** | **0.158** | **0.109** | **0.161** | **0.121** | **0.174** | **0.124** | **0.176** | **0.060** | **0.104** | **0.060** | **0.104** |
| Real | Transformer (T = 96) | 0.105 | 0.162 | 0.108 | 0.165 | 0.093 | 0.143 | 0.096 | 0.147 | 0.115 | 0.175 | 0.117 | 0.176 | 0.059 | 0.105 | 0.059 | 0.104 |
| GMEL | | 0.186 | 0.256 | 0.180 | 0.240 | 0.218 | 0.276 | 0.232 | 0.294 | 0.231 | 0.281 | 0.236 | 0.286 | 0.091 | 0.141 | 0.145 | 0.184 |
| DFG | | 0.341 | 0.467 | 0.323 | 0.442 | 0.175 | 0.237 | 0.168 | 0.228 | 0.354 | 0.467 | 0.360 | 0.472 | 0.068 | 0.116 | 0.069 | 0.117 |
| KSTDiff | | 0.387 | 0.526 | 0.397 | 0.538 | 0.420 | 0.546 | 0.492 | 0.616 | 0.448 | 0.564 | 0.484 | 0.604 | 0.243 | 0.386 | 0.236 | 0.375 |
| CGAN | | 0.195 | 0.268 | 0.197 | 0.267 | 0.362 | 0.447 | 0.385 | 0.476 | 0.300 | 0.388 | 0.296 | 0.380 | 0.124 | 0.175 | 0.133 | 0.187 |
| Diffwave | | 0.240 | 0.297 | 0.235 | 0.304 | 0.272 | 0.320 | 0.274 | 0.319 | 0.294 | 0.342 | 0.297 | 0.348 | 0.144 | 0.209 | 0.139 | 0.205 |
| DiT | | 0.231 | 0.287 | 0.227 | 0.280 | 0.285 | 0.327 | 0.278 | 0.321 | 0.308 | 0.346 | 0.297 | 0.339 | 0.146 | 0.208 | 0.151 | 0.208 |
| DDPM | | 0.114 | 0.169 | 0.118 | 0.173 | 0.122 | 0.175 | 0.121 | 0.176 | 0.130 | 0.183 | 0.138 | 0.191 | 0.070 | 0.116 | 0.071 | 0.115 |
| CVAE | | 0.153 | 0.214 | 0.158 | 0.219 | 0.255 | 0.332 | 0.261 | 0.336 | 0.170 | 0.230 | 0.174 | 0.231 | 0.113 | 0.169 | 0.117 | 0.169 |
| **CRAFT** | | **0.106** | **0.160** | **0.108** | **0.162** | **0.104** | **0.150** | **0.108** | **0.158** | **0.122** | **0.178** | **0.128** | **0.183** | **0.063** | **0.109** | **0.064** | **0.108** |
| Real | Transformer (T = 120) | 0.100 | 0.155 | 0.103 | 0.159 | 0.090 | 0.140 | 0.093 | 0.144 | 0.116 | 0.175 | 0.118 | 0.177 | 0.057 | 0.105 | 0.057 | 0.105 |
| GMEL | | 0.192 | 0.256 | 0.193 | 0.261 | 0.222 | 0.290 | 0.232 | 0.304 | 0.249 | 0.305 | 0.250 | 0.306 | 0.094 | 0.150 | 0.098 | 0.156 |
| DFG | | 0.310 | 0.425 | 0.303 | 0.413 | 0.150 | 0.203 | 0.151 | 0.208 | 0.349 | 0.460 | 0.354 | 0.465 | 0.071 | 0.122 | 0.072 | 0.123 |
| KSTDiff | | 0.390 | 0.436 | 0.621 | 0.699 | 0.509 | 0.654 | 0.507 | 0.640 | 0.328 | 0.380 | 0.329 | 0.386 | 0.721 | 0.771 | 0.341 | 0.420 |
| CGAN | | 0.270 | 0.369 | 0.282 | 0.390 | 0.283 | 0.375 | 0.288 | 0.381 | 0.273 | 0.349 | 0.261 | 0.338 | 0.108 | 0.175 | 0.109 | 0.173 |
| Diffwave | | 0.221 | 0.281 | 0.222 | 0.284 | 0.270 | 0.342 | 0.262 | 0.321 | 0.309 | 0.350 | 0.294 | 0.343 | 0.140 | 0.207 | 0.137 | 0.202 |
| DiT | | 0.222 | 0.280 | 0.222 | 0.276 | 0.265 | 0.318 | 0.268 | 0.315 | 0.306 | 0.349 | 0.303 | 0.347 | 0.173 | 0.220 | 0.149 | 0.208 |
| DDPM | | 0.109 | **0.161** | 0.113 | 0.166 | 0.112 | 0.163 | 0.113 | 0.166 | 0.129 | 0.184 | 0.135 | 0.190 | 0.067 | 0.111 | 0.066 | 0.109 |
| CVAE | | 0.153 | 0.214 | 0.158 | 0.220 | 0.202 | 0.272 | 0.212 | 0.283 | 0.170 | 0.231 | 0.173 | 0.231 | 0.106 | 0.171 | 0.105 | 0.168 |
| **CRAFT** | | **0.107** | 0.162 | **0.109** | **0.163** | **0.103** | **0.152** | **0.107** | **0.156** | **0.120** | **0.173** | **0.123** | **0.177** | **0.063** | **0.104** | **0.063** | **0.104** |
| Real | Transformer (T = 144) | 0.101 | 0.155 | 0.104 | 0.159 | 0.091 | 0.140 | 0.093 | 0.144 | 0.118 | 0.177 | 0.120 | 0.180 | 0.055 | 0.104 | 0.056 | 0.104 |
| GMEL | | 0.211 | 0.281 | 0.209 | 0.285 | 0.242 | 0.301 | 0.234 | 0.284 | 0.264 | 0.317 | 0.263 | 0.312 | 0.088 | 0.140 | 0.090 | 0.142 |
| DFG | | 0.323 | 0.450 | 0.307 | 0.427 | 0.188 | 0.248 | 0.185 | 0.247 | 0.399 | 0.521 | 0.391 | 0.510 | 0.069 | 0.120 | 0.069 | 0.119 |
| KSTDiff | | 0.524 | 0.586 | 0.459 | 0.510 | 0.488 | 0.618 | 0.369 | 0.423 | 0.367 | 0.462 | 0.385 | 0.446 | 0.190 | 0.315 | 0.166 | 0.239 |
| CGAN | | 0.342 | 0.431 | 0.392 | 0.480 | 0.371 | 0.463 | 0.352 | 0.449 | 0.284 | 0.360 | 0.288 | 0.368 | 0.133 | 0.186 | 0.135 | 0.184 |
| Diffwave | | 0.221 | 0.280 | 0.215 | 0.273 | 0.291 | 0.334 | 0.296 | 0.353 | 0.290 | 0.357 | 0.288 | 0.359 | 0.139 | 0.205 | 0.136 | 0.202 |
| DiT | | 0.223 | 0.279 | 0.221 | 0.276 | 0.290 | 0.331 | 0.283 | 0.322 | 0.297 | 0.338 | 0.301 | 0.342 | 0.147 | 0.208 | 0.143 | 0.203 |
| DDPM | | 0.113 | 0.167 | 0.117 | 0.173 | 0.122 | 0.173 | 0.124 | 0.176 | 0.127 | 0.181 | 0.131 | 0.185 | 0.071 | 0.113 | 0.069 | 0.110 |
| CVAE | | 0.192 | 0.260 | 0.193 | 0.261 | 0.277 | 0.359 | 0.283 | 0.363 | 0.234 | 0.297 | 0.240 | 0.305 | 0.124 | 0.198 | 0.124 | 0.196 |
| **CRAFT** | | **0.107** | **0.160** | **0.110** | **0.163** | **0.106** | **0.154** | **0.111** | **0.159** | **0.125** | **0.180** | **0.129** | **0.183** | **0.066** | **0.107** | **0.065** | **0.105** |
| Real | Transformer (T = 168) | 0.104 | 0.160 | 0.108 | 0.164 | 0.090 | 0.140 | 0.093 | 0.144 | 0.115 | 0.173 | 0.118 | 0.176 | 0.054 | 0.102 | 0.055 | 0.103 |
| GMEL | | 0.216 | 0.288 | 0.217 | 0.287 | 0.219 | 0.281 | 0.218 | 0.277 | 0.238 | 0.301 | 0.241 | 0.297 | 0.093 | 0.148 | 0.095 | 0.150 |
| DFG | | 0.315 | 0.427 | 0.313 | 0.424 | 0.162 | 0.217 | 0.167 | 0.226 | 0.377 | 0.493 | 0.374 | 0.492 | 0.071 | 0.122 | 0.072 | 0.123 |
| KSTDiff | | 0.344 | 0.473 | 0.340 | 0.467 | 0.444 | 0.526 | 0.510 | 0.625 | 0.395 | 0.508 | 0.374 | 0.473 | 0.199 | 0.339 | 0.196 | 0.308 |
| CGAN | | 0.385 | 0.469 | 0.385 | 0.476 | 0.324 | 0.418 | 0.323 | 0.412 | 0.319 | 0.385 | 0.346 | 0.413 | 0.140 | 0.200 | 0.145 | 0.210 |
| Diffwave | | 0.228 | 0.288 | 0.221 | 0.283 | 0.281 | 0.328 | 0.267 | 0.321 | 0.290 | 0.339 | 0.289 | 0.341 | 0.142 | 0.206 | 0.136 | 0.202 |
| DiT | | 0.231 | 0.288 | 0.227 | 0.282 | 0.283 | 0.325 | 0.267 | 0.318 | 0.305 | 0.342 | 0.308 | 0.347 | 0.192 | 0.231 | 0.154 | 0.206 |
| DDPM | | 0.115 | 0.169 | 0.119 | 0.174 | 0.129 | 0.178 | 0.132 | 0.182 | 0.127 | 0.178 | 0.136 | 0.187 | 0.071 | **0.112** | 0.069 | **0.110** |
| CVAE | | 0.193 | 0.257 | 0.195 | 0.259 | 0.285 | 0.369 | 0.287 | 0.370 | 0.196 | 0.247 | 0.202 | 0.252 | 0.138 | 0.189 | 0.138 | 0.188 |
| **CRAFT** | | **0.106** | **0.158** | **0.108** | **0.162** | **0.102** | **0.149** | **0.106** | **0.154** | **0.122** | **0.176** | **0.125** | **0.179** | **0.069** | 0.114 | **0.067** | 0.111 |

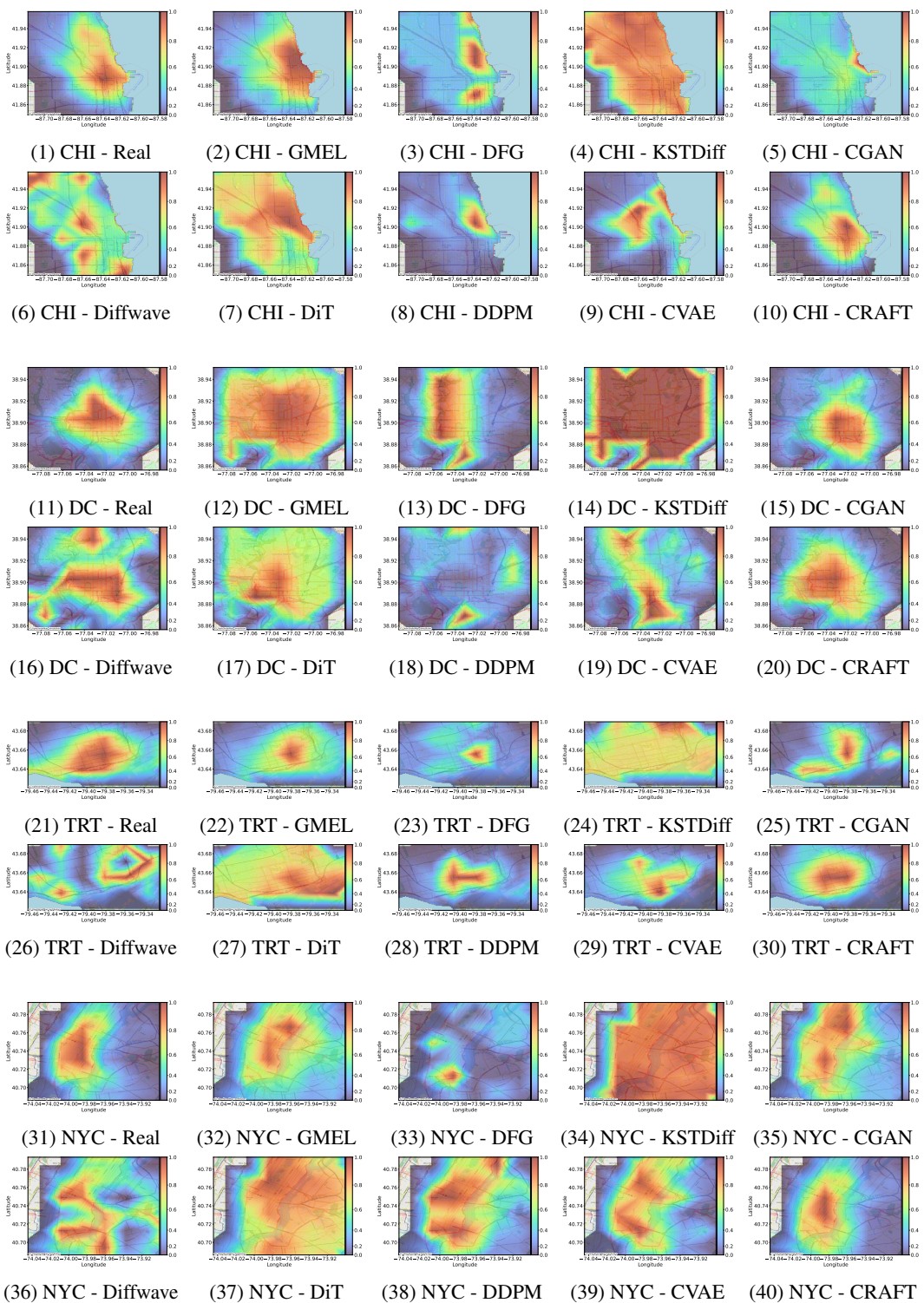

Figure 11: Heatmap of Traffic Flow in Different Cities (CHI stands for Chicago, DC stands for Washington, D.C., TRT stands for Toronto and NYC stands for New York City)

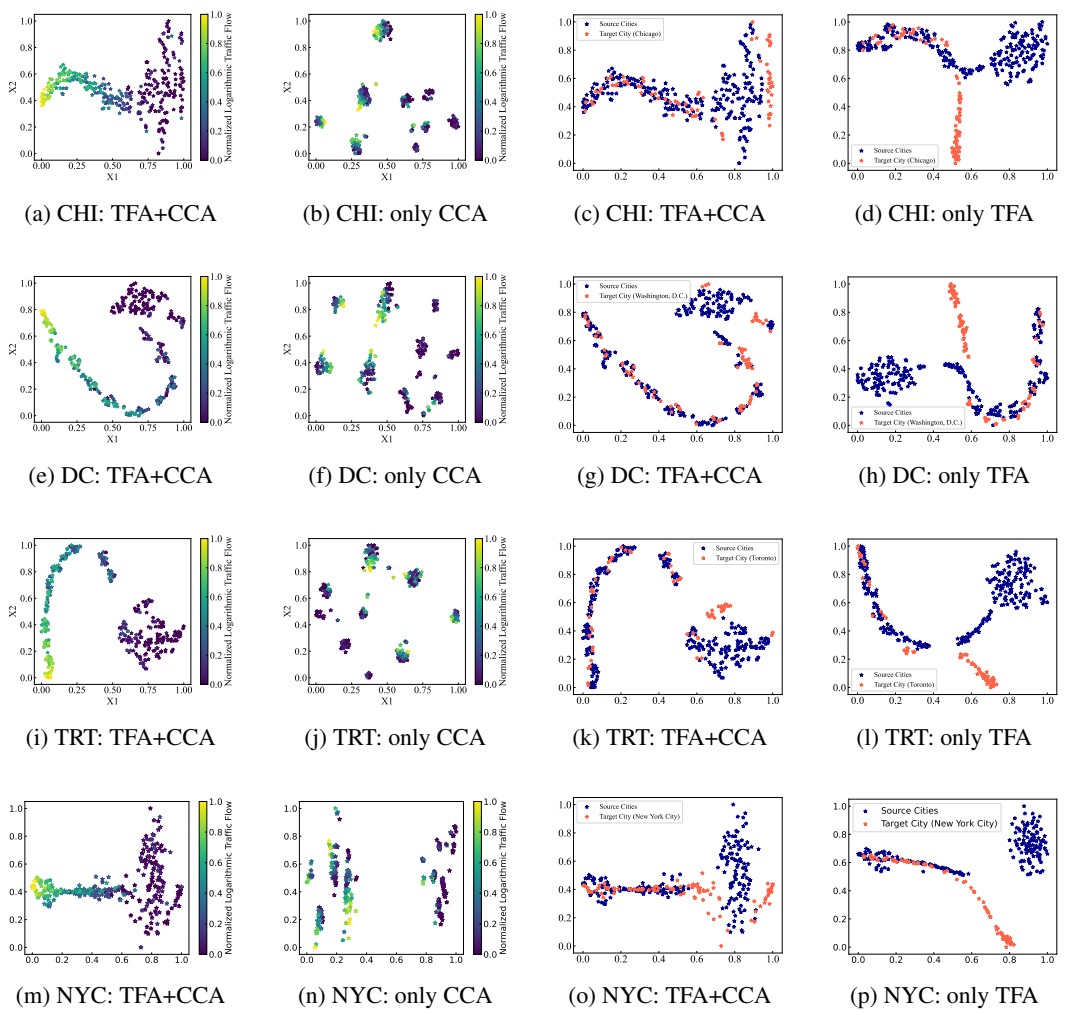

Figure 12: Visualization analysis for TFA and CCA (CHI stands for Chicago, DC stands for Washington, D.C., TRT stands for Toronto and NYC stands for New York City)

