# OpenReview forum: "Cross City Traffic Flow Generation via Retrieval Augmented Diffusion Model"
_NeurIPS.cc/2025/Conference — NeurIPS 2025 poster_

### Official Review · Reviewer_qp6Z · 2025-06-23

**Clarity:** 4
**Significance:** 3
**Originality:** 3
**Rating:** 5
**Confidence:** 4

**Summary:**

This work proposes CRAFT, a retrieval-augmented diffusion model designed for cross-city traffic flow generation. CRAFT addresses the domain shift problem across cities through Geographic Feature Alignmentand supplements missing dynamic traffic patterns in target cities using Retrieval-augmented Conditional generation. This enables the model to generate high-quality traffic flow data under zero-shot settings. Experimental results demonstrate that CRAFT significantly outperforms existing methods on four real-world city datasets, and the generated data achieves performance close to real data in downstream tasks, offering a practical solution for cities lacking historical traffic data.

**Questions:**

1. Can population, road network, and POIs fully capture a region’s traffic flow patterns? In real-world road networks, traffic patterns are also dynamically influenced by neighboring regions. Does the proposed model account for such inter-regional dependencies?
2. In the temporal length experiments, why does the baseline DDPM method exhibit improved performance as T increases? Can this scaling behavior be explained? Since the proposed method also adopts DDPM as its backbone, why does it not demonstrate similar scaling capabilities?
3. Why is the evaluation restricted to bike-sharing grid data? Were other grid-based datasets considered to ensure broader applicability?

**Ethical Concerns:**

["NO or VERY MINOR ethics concerns only"]

**Final Justification:**

During the rebuttal process, the authors have largely addressed the concerns I raised, including the following key points:

1. **Concerns about the inference efficiency of the diffusion model**: The authors evaluated different sequence lengths and data proportions using various diffusion model backbones, demonstrating strong inference performance.
2. **Curiosity about broader dataset domains**: In addition to the bike-related dataset, they incorporated two datasets based on Traffic4Cast, achieving state-of-the-art (SOTA) model performance.
3. **Impact of generation length on the model**: The authors clarified the specific meaning of "length" and explained how model performance varies with different input and generation lengths.
4. **Other issues**: The authors also elaborated on the key differences from DDPM and the motivation behind certain feature selections.

Overall, the authors provided detailed responses to my questions and outlined plans for future improvements. I believe their work is worthy of acceptance and look forward to seeing the enhancements in the final version.

**Limitations:**

Yes.

**Paper Formatting Concerns:**

None.

**Quality:**

3

**Strengths And Weaknesses:**

Strengths:
1. The contrastive learning approach effectively distinguishes different traffic patterns across regions, facilitating the learning of a unified and transferable region-traffic pattern representation, which benefits downstream traffic flow generation tasks.
2. The experimental results demonstrate strong performance in cross-city traffic flow generation and subsequent prediction tasks, validating the effectiveness of the proposed traffic pattern alignment and cross-city alignment strategies.

Weaknesses:
1. Using a diffusion model as the backbone of the framework may impact efficiency, yet there is no evaluation or comparison of training/inference time or computational costs.
2. The experiments are limited to bike-sharing datasets, lacking robustness validation across broader transportation domains (e.g., vehicular traffic, pedestrian flow).

---

> ### Author Rebuttal · Authors · 2025-07-31
>
> Thanks so much for your detailed and constructive comments. Please see our below responses to your concerns one by one.
>
> # W1:
>
> **1) Computational costs**
>
> To evaluate computational cost, we tested the model’s inference time on progressively larger portions of the data—20%, 40%, 60%, 80%, and 100%—as shown in Table below.
>
> |Data portation|Train time per epoch|Valid time per epoch|Avg memory (GB)|Peak memory (GB)|
> |:-:|:-:|:-:|:-:|:-:|
> |0.20|7.362|1.032|0.343|0.553|
> |0.40|15.383|2.223|0.345|0.553|
> |0.60|24.294|3.177|0.354|0.553|
> |0.80|31.069|4.263|0.347|0.553|
> |1.00|39.687|5.294|0.345|0.553|
>
> Next, with the data scale fixed, we gradually increased the sequence length and measured inference time. Results are reported in Table below.
>
> |Seq leangth|Model size (Byte)|Train time per epoch|Valid time per epcoh|Avg memory (GB)|
> |:-:|:-:|:-:|:-:|:-:|
> |24|51781096|40.999|5.685|0.343|
> |48|51830248|41.156|5.531|0.344|
> |72|51879400|43.729|6.053|0.344|
> |96|51928552|44.555|5.631|0.345|
> |120|51977704|46.712|6.429|0.345|
> |144|52026856|48.883|6.666|0.345|
> |168|52076008|51.896|6.351|0.346|
>
> Combining the above tables, we observe that our model's computational cost grows linearly with data scale and is only slightly affected by sequence length. Memory cost is also insensitive to both data scale and sequence length.
>
> **2) Time complexity and efficiency**
>
> In below table, we report both inference and training times for CRAFT and the baselines. CRAFT includes two variants: CRAFT-DDPM and CRAFT-DDIM. As expected, diffusion-based models are generally slower in both training and inference than single-shot generative models. CRAFT-DDPM achieves comparable efficiency to other diffusion-based models such as DiT but remains slower than single-shot models like GMEL. CRAFT-DDIM improves the sampling strategy of the diffusion backbone, achieving efficiency comparable to, or even surpassing, some single-shot models (e.g., DFG).
>
> |Method|Infer time(s)|Infer speed(samples/sec)|Training time(s)|
> |:-:|:-:|:-:|:-:|
> |GMEL|23.84|1701.63|5639|
> |CVAE|14.20|2883.65|2307|
> |CGAN|10.75|3775.23|12144|
> |DFG|46.54|871.63|36077|
> |KSTDiff|18003.47|2.13|73006|
> |Diffwave|420.83|96.40|3778|
> |DiT|1923.11|21.10|12430|
> |DDPM|1925.93|21.06|13024|
> |CRAFT-DDPM|1926.43|21.06|13492|
> |CRAFT-DDIM|21.08|1924.65|13492|
>
> In below table, we compare the performance of CRAFT-DDPM and CRAFT-DDIM. CRAFT-DDIM achieves performance close to CRAFT-DDPM and outperforms all other baselines. Therefore, for optimal performance, CRAFT-DDPM is preferred, while CRAFT-DDIM offers a better trade-off between performance and efficiency.
>
> |Target City|Method||Inflow|||Outflow||
> |:-:|:-:|:-:|:-:|:-:|:-:|:-:|:-:|
> |||CPC|NMAE|NRMSE|CPC|NMAE|NRMSE|
> |CHI|CRAFT-DDPM|0.785|0.140|0.216|0.786|0.140|0.216|
> ||CRAFT-DDIM|0.778|0.146|0.225|0.761|0.151|0.236|
> |DC|CRAFT-DDPM|0.815|0.158|0.240|0.816|0.159|0.240|
> ||CRAFT-DDIM|0.791|0.172|0.253|0.788|0.176|0.257|
> |Toronto|CRAFT-DDPM|0.804|0.178|0.267|0.804|0.179|0.268|
> ||CRAFT-DDIM|0.803|0.181|0.293|0.801|0.182|0.295|
> |NYC|CRAFT-DDPM|0.782|0.103|0.170|0.786|0.102|0.165|
> ||CRAFT-DDIM|0.778|0.196|0.186|0.734|0.105|0.169|
>
> # W2 & Q3: Can CRAFT generalize to more diverse domains?
>
> We have added experiments on the Traffic4Cast[1] dataset, which contains vehicle flow spans diverse countries across Europe and Asia and exhibits greater inter-city variation. In addition, Traffic4Cast offers larger-scale and finer-grained data than the datasets used in our main paper. Specifically, we selected Moscow and Berlin to evaluate CRAFT. The results are shown in the table below.
>
> |Target City|Method||Inflow|||Outflow||
> |:-:|:-:|:-:|:-:|:-:|:-:|:-:|:-:|
> |||CPC|NMAE|NRMSE|CPC|NMAE|NRMSE|
> |Berlin|GMEL|0.500|0.442|0.501|0.487|0.470|0.528|
> ||DiT|0.436|0.353|0.444|0.453|0.494|0.571|
> ||Diffwave|0.424|0.562|0.662|0.386|0.710|0.768|
> ||DDPM|0.388|0.333|0.395|0.389|0.342|0.414|
> ||CRAFT|**0.537**|**0.312**|**0.381**|**0.537**|**0.313**|**0.381**|
> |Moscow|GMEL|0.182|0.357|0.431|0.254|0.345|0.418|
> ||DiT|0.566|0.403|0.480|0.500|0.320|0.396|
> ||Diffwave|0.170|0.397|0.469|0.486|0.530|0.594|
> ||DDPM|0.338|0.334|0.409|0.334|0.334|0.409|
> ||CRAFT|**0.609**|**0.239**|**0.315**|**0.607**|**0.240**|**0.316**|
>
> The experimental results show that CRAFT not only generalizes well to the traffic4cast dataset but also achieves larger improvements over baselines compared to our original datasets, highlighting its strong generalization capability. We sincerely thank you for the suggestion and will include these results in the revised version.
>
> [1] NeurIPS traffic4cast 2021
>
> # Q1：
>
> **1) Can CRAFT model inter-region relationships?**
>
> CRAFT has considered inter-region relationships. Specifically, our spatial encoder, a graph transformer, models the inter-region relationships among neighboring regions when representing geographic features.
>
> **2) Why does CRAFT focus on basic features such as POIs, roads, and population?**
>
> Flow magnitude is primarily determined by population size. Flow patterns are shaped by regional functionality which is indicated by the types and amounts of POIs and roads. In our original paper, to ensure the cross-city generalizability, we selelct only the most basic spatiotemporal features shared across all cities. Since cities share only a few fundamental geographic features and most other features differ significantly, incorporating too many features harms geographic feature alignment. This is a key challenge in cross-city flow generation: to reduce significant domain shifts, the generation must depend on a limited feature set.
>
> # Q2: How does the length of T affect the DDPM’s performance?
>
> **1) DDPM’s performance fluctuates with varying T**
> As shown in paper’s Figure 7, DDPM’s performance varies within a narrow ±0.1 range as T increases. On the NY dataset, performance slightly decreases with larger T, while on the D.C. dataset, a clear improvement is observed. Among all baselines, DDPM exhibits the most stable performance trend.
>
> **2) Why does this fluctuation occur?**
> How the performance variation with increasing T depends on the distribution of the target data. As T represents both the input sequence length (T_input) and the output sequence length (T_output). When T = 24, it denotes both input length and output length are 24. Thus, increasing T simultaneously increases task difficulty (due to longer output length per sample) but provides richer information per input sample (due to longer input length).
>
> Specifically, detailed analysis can be shown in the below table, where we select Chicago as the target city and evaluate using NMAE (lower is better). We test various combinations of input and output lengths. For example, 24→48 denotes an input length of 24 and an output length of 48.
>
> ||NMAE||NMAE|
> |:-:|:-:|:-:|:-:|
> |24->48|0.150|48->48|0.146|
> |24->72|0.154|72->72|0.147|
> |24->96|0.156|96->96|0.148|
> |24->120|0.158|120->120|0.146|
> |24->144|0.159|144->144|0.148|
>
> Within each row, with a fixed output length, model performance improves as input length increases; Within each column, with a fixed input length, performance degrades as output length increases.
>
> Task difficulty is closely tied to the variance of the target data: higher variance—caused by larger fluctuations—leads to more challenging generation, while lower variance indicates easier generation. We computed the target variance across the four datasets, as shown in the table below：
>
> |NY|Chicago|Toronto|DC|
> |:-:|:-:|:-:|:-:|
> |0.227|0.226|0.220|0.215|
>
> As illustrated, the NY dataset has the highest variance, making long-sequence generation most difficult. In contrast, the D.C. dataset shows the lowest variance, making the task relatively easier.
>
> Therefore, when the target data exhibits high variance, the loss from a larger output length dominates, leading to performance degradation as T increases; conversely, when the variance is low, the gain from a longer input length prevails, resulting in improved performance with increasing T. This aligns with the results in Table 7 of the paper.
>
> **3) Why is CRAFT different from DDPM?**
>
> Regardless of the length of T, CRAFT’s RCA module consistently retrieves relevant historical data from source cities as reference, effectively smoothing fluctuations and enhancing input quality. As a result, CRAFT performs more stably and achieves superior results in most cases.

---

> > ### Comment · Reviewer_qp6Z · 2025-08-04
> > **Response to Rebuttal**
> >
> > Thank you to the authors for your rebuttal. The detailed experiments provided by the authors have addressed my concerns.
> >
> > Additional suggestions: Include more discussion on the model architecture, supported by experimental validation, to better articulate the paper’s main contributions and key findings.

---

> > > ### Author Response · Authors · 2025-08-04
> > >
> > > We sincerely thanks for your response and valuable suggestions.  Accordingly, we will add a discussion section comprising the following two components:
> > >
> > > **1.Comparison with Existing Baselines:**
> > >
> > > Current baseline methods can be categorized into two groups:
> > >
> > > - Deterministic predictive models：such as GMEL and DFG, adopt a naive representation module (e.g., GAT or DCN) with an MLP prediction head. These models usually lacks the ability to model data uncertainty.
> > >
> > > - Conditional generative models：Such as GANs, VAEs, and diffusion models offer improved capacity for modeling uncertainty and dynamic behaviors. However, most existing methods in this category do not incorporate cross-city contextual conditions, making them less effective in domain-shift scenarios such as cross-city transfer.
> > >
> > > In contrast, CRAFT is a diffusion-based framework，which is explicitly designed to answer how to incorporate cross-city conditions:
> > >
> > > - GFA: a graph transformer-based encoder that enables adaptive geographic feature alignment across cities.
> > >
> > > - RCA: a retrieval-augmented mechanism to answer how to involve dynamic references under zero-shot settings.
> > >
> > > **2. Additional experiments of CRAFT's Applicability:**
> > >
> > > - Efficiency and Complexity: we will summarize our response to the weakness 1 with its supporting experiments to discuss of CRAFT's efficiency and complexity costs.
> > >
> > > - Generalization to Broader Data: we will summarize our experiments to weakness 2 and question 3. We will extend our experiments to more diverse datasets and present a case study demonstrating CRAFT's generalization ability.
> > >
> > >
> > > We sincerely thanks for your valuable suggestions and for recognizing the contributions of our work.
> > >
> > > We hope that our responses addressed your concerns. We would like to kindly ask if, you find our responses satisfactory, could you possibly reconsider and adjust the final score. We would be deeply grateful for your reconsideration.
> > >
> > > Regardless, we always fully respect your final decision and sincerely appreciate the time and patience you have devoted to reviewing our paper.
> > >
> > > Sincerely,
> > >
> > > The Authors

---

### Official Review · Reviewer_vAPG · 2025-06-23

**Clarity:** 3
**Significance:** 3
**Originality:** 3
**Rating:** 4
**Confidence:** 4

**Summary:**

This paper proposes CRAFT, a retrieval-augmented diffusion framework for zero-shot cross-city traffic flow generation. It aligns geographic features across source and target cities using both flow-based and optimal transport-based losses, augments conditions by retrieving similar temporal patterns from source cities, and generates region-level inflow and outflow sequences from noise. Experiments on four North American cities with a leave-one-city-out protocol show that CRAFT outperforms multiple generative baselines in generation quality and downstream forecasting utility.

**Questions:**

- It seems that the current implementation only considers feature-based alignments. Would incorporating explicit road network topology or intersection connectivity further improve alignment?
- Can the model capture extreme events such as holiday surges or incident-driven congestion?
- What are the computational and memory costs when scaling to larger source datasets?

**Ethical Concerns:**

["NO or VERY MINOR ethics concerns only"]

**Final Justification:**

The authors provide sufficient evidence to allow me to adjust my score.

**Limitations:**

While CRAFT establishes a promising framework for zero-shot cross-city flow generation, it could benefit from integrating richer road network topology and dynamic event indicators to better capture heterogeneous congestion patterns and unusual traffic spikes. Expanding the evaluation to include cities with non-grid layouts and providing visualizations of retrieved temporal snippets would clarify the retrieval module’s impact and support more informed tuning of retrieval strategies.

**Quality:**

3

**Strengths And Weaknesses:**

### Strengths
- Introduces a novel zero-shot cross-city generation task with clear practical applications for data-scarce cities
- Combines geographic alignment and retrieval modules within a conditional diffusion model in an end-to-end trainable framework
- Provides comprehensive evaluation across multiple cities and assesses both generation metrics and downstream prediction performance

### Weaknesses
- The method mainly integrates existing techniques such as graph-based geographic encoding, retrieval-augmented conditioning, and conditional diffusion models, without introducing novel algorithmic components or architecture-level innovations
- Lacks detailed analysis or visualization of the retrieved temporal patterns and their contribution
- Evaluation is limited to four similar grid-based North American cities and does not cover diverse urban layouts or special event scenarios

---

> ### Author Rebuttal · Authors · 2025-07-31
>
> Thanks so much for your detailed and constructive comments. Please see our below responses to your concerns one by one.
>
> # W1: The novelty and contributions of our model
>
> Cross-city flow generation is an emerging area with limited existing efforts. This is because flow generation itself is a challenging problem—it requires geographic features as input to map static spatial structures to dynamic flows. Moreover, the cross-city setting further requires the model to handle domain shifts in both static geographic features and dynamic flow patterns across different cities, even in a completely zero-shot manner.
>
> Therefore, this area remains in its early stages and is still under explored. Existing works primarily focus on specific types of data, such as origin-destination (OD) data[1] or cellular signal data[2], which often come with type-specific priors that simplify the task.
>
> In contrast, we targets generation on general flow types which is under-explored, and address two critical challenges: “severe domain shift” and “insufficient conditions”.
>
> + To address domain shift, we were among the early efforts to introduce optimal transport theory into flow tasks and proposed the GFA module. Compared with prior alignment approaches, GFA adaptively selects the most relevant source regions for each target region, enabling finer-grained alignment.
>
> + To address insufficient conditions, we adapted RAG for flow generation and developed the RCA module, which integrates dynamic flow information from source cities under zero-shot settings. Most prior works relied solely on static geographic features. RCA allows our model to adapt to more complex flow patterns, resulting in more accurate flow generation.
>
> [1]A Large-scale Benchmark Dataset for Commuting Origin-destination Matrix Generation
>
> [2]Deep Transfer Learning for City-scale Cellular Traffic Generation through Urban Knowledge Graph
>
> # W2 & Limitation 2: Detailed analysis of the retrieval augmented module (RCA)
>
> For a given target region and timestamp, RCA retrieves the most similar historical flow segments from source cities based on two features: geographic similarity and temporal proximity.
>
> We conducted ablations on each feature to assess their individual contributions. The results are shown in the table below. Specifically, we averaged the flow segments retrieved by RCA and measured their similarity to the ground-truth flow in the target city using Pearson, Spearman, and Kendall metrics.
>
> |Target City|Method||Inflow|||Outflow||
> |:-:|:-:|:-:|:-:|:-:|:-:|:-:|:-:|
> |||pearson|spearman|kendall|pearson|spearman|kendall|
> |CHI|RCA w/o time|0.330|0.314|0.215|0.330|0.316|0.217|
> ||RCA w/o geo|0.606|0.692|0.507|0.599|0.680|0.495|
> ||RCA|0.830|0.840|0.652|0.826|0.847|0.660|
> |DC|RCA w/o time|0.280|0.270|0.188|0.268|0.249|0.172|
> ||RCA w/o geo|0.573|0.624|0.451|0.575|0.629|0.453|
> ||RCA|0.792|0.832|0.644|0.767|0.813|0.622|
> |Toronto|RCA w/o time|0.205|0.207|0.143|0.210|0.214|0.147|
> ||RCA w/o geo|0.707|0.737|0.549|0.700|0.726|0.541|
> ||RCA|0.770|0.787|0.603|0.763|0.788|0.606|
> |NYC|RCA w/o time|0.484|0.476|0.329|0.480|0.458|0.318|
> ||RCA w/o geo|0.467|0.571|0.403|0.467|0.579|0.409|
> ||RCA|0.865|0.845|0.657|0.867|0.848|0.660|
>
> Both geo and temporal features contribute significantly to performance improvements, with temporal proximity often playing a more dominant role. This indicates that flow patterns across regions are more strongly influenced by temporal periodicity, while also being shaped by static geographic features.
>
> # W3 & Limitation 2:
>
> **1) Can CRAFT utilize non-grid data?**
>
> Yes, actually datasets used in our paper is in graph format, where each node is a grid. CRAFT is not limited to grid-based data. It only requires that each basic geographic unit—regardless of its shape—provides complete information of POIs, roads, and population. In practice, such units are usually implemented as grids.
>
> **2) Can CRAFT generalize to more diverse cities?**
>
> We have added experiments on the Traffic4Cast[3] dataset, which contains vehicle flow spans diverse countries across Europe and Asia and exhibits greater inter-city variation. In addition, Traffic4Cast offers larger-scale and finer-grained data than the datasets used in our main paper. Specifically, we selected Moscow and Berlin to evaluate CRAFT. The results are shown in the table below.
>
> |Target City|Method||Inflow|||Outflow||
> |:-:|:-:|:-:|:-:|:-:|:-:|:-:|:-:|
> |||CPC|NMAE|NRMSE|CPC|NMAE|NRMSE|
> |Berlin|GMEL|0.500|0.442|0.501|0.487|0.470|0.528|
> ||DiT|0.436|0.353|0.444|0.453|0.494|0.571|
> ||Diffwave|0.424|0.562|0.662|0.386|0.710|0.768|
> ||DDPM|0.388|0.333|0.395|0.389|0.342|0.414|
> ||CRAFT|**0.537**|**0.312**|**0.381**|**0.537**|**0.313**|**0.381**|
> |Moscow|GMEL|0.182|0.357|0.431|0.254|0.345|0.418|
> ||DiT|0.566|0.403|0.480|0.500|0.320|0.396|
> ||Diffwave|0.170|0.397|0.469|0.486|0.530|0.594|
> ||DDPM|0.338|0.334|0.409|0.334|0.334|0.409|
> ||CRAFT|**0.609**|**0.239**|**0.315**|**0.607**|**0.240**|**0.316**|
>
> The experimental results show that CRAFT not only generalizes well to the traffic4cast dataset but also achieves larger improvements over baselines compared to our original datasets, highlighting its strong generalization capability. We sincerely thank you for the suggestion and will include these results in the revised version.
>
> [3] NeurIPS traffic4cast 2021
>
> # Q1 & Limitation 1: Enriching geographical features via network topology and intersection connectivity
>
> Thank you for your comment. We computed each node’s space syntax features[4], including degree, closeness centrality, and betweenness centrality, to capture explicit road topology and intersection connectivity. Results are shown in the table below, where CRAFT + syntax denotes CRAFT augmented with explicit topology and intersection connections.
>
> |Target City|Method||Inflow|||Outflow||
> |:-:|:-:|:-:|:-:|:-:|:-:|:-:|:-:|
> |||CPC|NMAE|NRMSE|CPC|NMAE|NRMSE|
> |CHI|CRAFT|0.785|0.140|0.216|0.786|0.140|0.216|
> ||CRAFT+syntax|0.814|0.121|0.187|0.815|0.123|0.189|
> |DC|CRAFT|0.815|0.158|0.240|0.816|0.159|0.240|
> ||CRAFT+syntax|0.800|0.165|0.246|0.799|0.168|0.247|
> |Toronto|CRAFT|0.804|0.178|0.267|0.804|0.179|0.268|
> ||CRAFT+syntax|0.827|0.159|0.239|0.827|0.159|0.240|
> |NYC|CRAFT|0.782|0.103|0.170|0.786|0.102|0.165|
> ||CRAFT+syntax|0.779|0.124|0.194|0.775|0.113|0.188|
>
> Experimental results show that space syntax features provide certain gains on the CHI and Toronto datasets, but lead to slight performance drops on DC and NYC. This is because DC and NYC have the simplest and most complex road network structures, respectively, among the four datasets.
>
> Specifically, flow magnitude is primarily determined by population size, while flow patterns are shaped by regional functionality. When cities have similar road network complexity, incorporating syntax features improves performance. However, when complexity differs, such features may impede cross-city spatial alignment. In our original paper, to ensure generalizability, we used only the most basic spatiotemporal features shared across all cities.
>
> [4]Space Syntax. Environment and Planning B: Planning and design
>
> # Q2 & W3 & Limitation1: Can CRAFT handle sudden changes in flow patterns, such as a holiday surge?
>
> As existing public flow datasets lack information on incidents or extreme events. We constructed a dataset with sudden flow pattern changes based on the four datasets used in our paper.
>
> Specifically, we kept all weekday (Monday to Friday) data in the training set, while using 80% of weekend and holiday data as the test set and adding the remaining 20% back into training. Additionally, we included a 0-1 one-hot vector as an event code to distinguish weekdays from holidays. We evaluated the impact of this event indicator on the RCA module’s performance, which is evaluated by similarity to the ground-truth flow in the target city using CPC, NMAE, NRMSE metrics. Results are in the table below:
>
> |Target City|Method||Inflow|||Outflow||
> |:-:|:-:|:-:|:-:|:-:|:-:|:-:|:-:|
> |||CPC|NMAE|NRMSE|CPC|NMAE|NRMSE|
> |CHI|RCA w/o event indicator|0.768|0.146|0.218|0.763|0.150|0.221|
> ||RCA|0.800|0.123|0.190|0.802|0.123|0.189|
> |DC|RCA w/o event indicator|0.799|0.164|0.235|0.798|0.166|0.239|
> ||RCA|0.810|0.150|0.218|0.805|0.155|0.224|
> |Toronto|RCA w/o event indicator|0.779|0.198|0.295|0.773|0.205|0.299|
> ||RCA|0.783|0.188|0.279|0.788|0.185|0.275|
> |NYC|RCA w/o event indicator|0.769|0.103|0.167|0.764|0.107|0.171|
> ||RCA|0.789|0.092|0.149|0.790|0.093|0.149|
>
> Thank you for your detailed advice. The event indicator indeed improves the model’s ability to handle abrupt changes in flow patterns, such as holiday surges, thus enhancing CRAFT’s capacity for such scenarios. We will further extend this result and include it as a case study in the revised version.
>
> # Q3: The computational and memory costs when scaling to larger datasets
>
> To evaluate computational cost, we tested the model’s inference time on progressively larger portions of the data—20%, 40%, 60%, 80%, and 100%—as shown in Table below.
>
> |Data portation|Train time per epoch|Valid time per epoch|Avg memory (GB)|Peak memory (GB)|
> |:-:|:-:|:-:|:-:|:-:|
> |0.20|7.362|1.032|0.343|0.553|
> |0.40|15.383|2.223|0.345|0.553|
> |0.60|24.294|3.177|0.354|0.553|
> |0.80|31.069|4.263|0.347|0.553|
> |1.00|39.687|5.294|0.345|0.553|
>
> Next, with the data scale fixed, we gradually increased the sequence length and measured inference time. Results are reported in Table below.
>
> |Seq leangth|Model size (Byte)|Train time per epoch|Valid time per epcoh|Avg memory (GB)|
> |:-:|:-:|:-:|:-:|:-:|
> |24|51781096|40.999|5.685|0.343|
> |48|51830248|41.156|5.531|0.344|
> |72|51879400|43.729|6.053|0.344|
> |96|51928552|44.555|5.631|0.345|
> |120|51977704|46.712|6.429|0.345|
> |144|52026856|48.883|6.666|0.345|
> |168|52076008|51.896|6.351|0.346|
>
> Combining the above tables, we observe that our model's computational cost grows linearly with data scale and is only slightly affected by sequence length. Memory cost is also insensitive to both data scale and sequence length.

---

> > ### Comment · Reviewer_vAPG · 2025-08-04
> >
> > I think the authors provide sufficient evidence to allow me to adjust my score. Thanks for the effort.

---

### Official Review · Reviewer_6r4v · 2025-07-02

**Clarity:** 3
**Significance:** 3
**Originality:** 3
**Rating:** 4
**Confidence:** 3

**Summary:**

This paper proposes CRAFT (Cross-city Retrieval-Augmented traffic Flow generaTion), a retrieval-augmented diffusion model with geographic representation alignment, to address the challenge of generating traffic flow data for new, data-lacking cities. Unlike existing methods that rely on city-specific data and struggle with cross-city generalization, CRAFT leverages data from multiple source cities and incorporates two key modules: Geographic Feature Alignment (GFA) to tackle domain shift by aligning geographic representations across cities through traffic flow alignment (TFA) and cross-city alignment (CCA), and Retrieval-based Condition Augmentation (RCA) to mitigate insufficient conditions by retrieving relevant historical traffic patterns from source cities. Built on the DDPM backbone, the model is trained on source cities and deployed zero-shot on target cities. Experiments on four real-world bicycle traffic datasets (Chicago, Washington D.C., Toronto, New York City) demonstrate that CRAFT outperforms existing baselines in cross-city zero-shot generation, achieving superior performance on metrics like CPC, NMAE, and NRMSE, with generated data showing high utility in downstream traffic prediction tasks, closely approaching the performance of real data.

**Questions:**

1. Does the model in the paper compare with existing cross-city-specific methods? And has the paper conducted sufficient literature research on cross-city traffic flow generation?

2. What is the time complexity of the CRAFT method proposed in the paper? And is the training and inference process of this method efficient?

3. The paper doesn't explicitly state if CRAFT can be applied to the traffic4cast dataset. But considering CRAFT is designed for cross - city traffic flow generation using geographic features and retrieval - augmented diffusion to handle domain shift and insufficient conditions, it might be adaptable to challenging datasets like traffic4cast. Its generalizable and flexible key components (Geographic Feature Alignment and Retrieval - based Condition Augmentation) imply that with suitable adjustments for the specific characteristics and complexity of traffic4cast, application could be feasible, though further experiments are needed to verify this.

**Ethical Concerns:**

["NO or VERY MINOR ethics concerns only"]

**Final Justification:**

The author's rebuttal has addressed most of my concerns. Thus, I will keep my positive scores.

**Limitations:**

yes

**Quality:**

3

**Strengths And Weaknesses:**

The strengths of this paper lie in several key aspects. Firstly, it addresses the critical challenge of cross-city traffic flow generation, a relatively underexplored area, by proposing a novel transfer learning framework that enables zero-shot generation in unseen target cities without relying on their historical data . Secondly, the introduced Geographic Feature Alignment (GFA) and Retrieval-based Condition Augmentation (RCA) modules effectively tackle domain shift and insufficient condition issues, respectively, and these lightweight, plug-in components do not require altering the backbone architecture, showcasing the method's simplicity and practicality . Additionally, extensive experiments on four real-world urban datasets demonstrate the model's state-of-the-art zero-shot generation performance, with significant improvements over existing baselines, and its generated data exhibits high utility in downstream traffic prediction tasks, closely approaching the performance of real data . Moreover, the paper provides sufficient details on the methodology, experimental settings, and open access to code and datasets, ensuring reproducibility .

A notable limitation of the paper's methodology lies in its insufficient literature review regarding cross-city traffic flow generation and transfer learning. In the "Related Work" section, while it briefly categorizes traffic flow generation models into physics-based, static, and dynamic ones, and mentions retrieval-augmented generation (RAG) applications in time series, it lacks a comprehensive survey of existing studies specifically focusing on cross-city scenarios. Additionally, the discussion on transfer learning, which is central to enabling cross-city generalization, is underdeveloped—there is little engagement with relevant transfer learning frameworks or prior attempts to address domain shift in urban traffic flow generation. This inadequacy results in a failure to fully contextualize the proposed method within the broader landscape of cross-city transfer learning research, weakening the connection between the methodology and the core cross-city challenge it aims to solve.

Another major issue with the method is that it fails to compare with existing cross-city-specific methods. Instead of engaging with approaches specifically designed for cross-city scenarios, the paper merely lists a set of general generative models (such as GMEL, DFG, KSTDiff, etc.) as baselines without distinguishing those tailored for cross-city transfer. This lack of comparison with dedicated cross-city methods makes it difficult to fully assess the novelty and advancement of the proposed CRAFT in addressing the unique challenges of cross-city traffic flow generation.

---

> ### Author Rebuttal · Authors · 2025-07-30
>
> Thanks so much for your detailed and constructive comments. Please see our below responses to your concerns.
>
> # W1 & Q1:
>
> **1) Literature review of cross-city flow generation**
>
> Flow prediction and generation are different tasks. Flow prediction leverages historical flow data to forecast future flows, focusing on modeling temporal patterns. In contrast, flow generation uses geographic features as input, mapping static spatial structure to dynamic flows. The main challenge of cross-city prediction lies in flow pattern shifts in different cities. While Cross-city generation must simultaneously handle shifts in both static geographic features and dynamic flow patterns.
>
> Some pioneer works [1][2] have explored cross-city flow prediction, the area remains underdeveloped, with existing efforts limited to specific types of data and tasks. [1] is an early effort to cross-city generation in origin-destination (OD) data, building a large-scale OD dataset for pretraining a universal OD flow model. However, OD data is a specific flow type with fixed patterns and strong periodicity, making [1] limited to OD flow and hard to generalized to tasks with broader flow types. [2]specialized for cellular signal data, as cellular stations inherently have unique priors and clear semantic links to surrounding geographic entities. Based on this character, [2] constructs a universal knowledge graph to guide cross-city transfer. Yet, this strategy is difficult to extend to general flow scenarios, where such structured priors are often unavailable.
>
> Existing efforts focus on specific types of data and tasks, our work aims to explore more general scenarios which remain under explored . Our work may serve as an initial prob to such tasks.
>
> [1]A Large-scale Benchmark Dataset for Commuting Origin-destination Matrix Generation
>
> [2]Deep Transfer Learning for City-scale Cellular Traffic Generation through Urban Knowledge Graph
>
> **2) Transfer learning in cross-city flow tasks**
>
> Transfer learning in cross-city tasks mostly focused on few-shot flow prediction but rarely applied to zero-shot flow generation. Since most methods require certain amounts of target city data during training. Specifically, they can be categorized into below three paradigms
>
> - meta-learning: [3][4] use meta-learning to obtain good model initialization from source cities. However, it requires few-shot adaptation on target cities and struggle with zero-shot tasks. But we targets zero-shot generation without any fine-tuning on target cities.
>
> - adversarial training: [5][6] leverage a domain discriminator to enforce models only extract domain-shared features. But these methods usually suffer from limited generalization ability as training discriminators is hard and typically assumes distribution gaps between domains are not too large.
>
> - feature alignment: uses an alignment loss to project source and target data into a shared feature space. Additionally, these methods typically employ feature decoupling to simplify alignment. [7], [8] decouple features along spatial-temporal dimensions, geographic structures, respectively. However, such methods are sensitive to hyper-parameters if alignment losses are too complex.
>
> In conclusion, transfer learning can address few-shot flow prediction in cross-city scenarios, they cannot fully solve the zero-shot flow generation, as original meta-learning and adversarial training still require access to flow data of target cities during training. But transfer learning can partially support feature transfer for flow generation. Inspired by the feature alignment paradigm, we designed the Geographic Feature Alignment(GFA) module to align the static geographic features of the source and target cities.
>
> [3]Cross-City Multi-Granular Adaptive Transfer Learning for Traffic Flow Prediction
>
> [4]Spatio-Temporal Meta-Graph Learning for Traffic Forecasting
>
> [5]Domain-Adversarial Training With Knowledge Transfer for Spatio-Temporal Prediction Across Cities
>
> [6]Generating Origin-Destination Flow for New Cities Via Domain Adversarial Training
>
> [7]Transferable Graph Structure Learning for Graph-based Traffic Forecasting Across Cities
>
> [8]Exploiting Hierarchical Correlations for Cross-City Cross-Mode Traffic Flow Prediction
>
> # W2 & Q1: Why most baselines are general generative models & comparisons with task-specific baselines
>
> As discussed in above literature review, cross-city flow generation is an emerging area with few baselines, most of which focus on specific data types. Thus, we selected general generative models as our baselines in the original paper.
>
> Inspired by your comments on transfer-learning, we incorporated transfer learning techniques to enhance the general baseline, adapting it more "task-specific". These improvements are applied to the alignment of static geographical features between source and target cities.
>
> Specifically, we applied two feature alignment losses:TCA and MMD, and an adapted adversarial training strategy(ADV). Experimental results are as follows:
>
> |Target City|Method||Inflow|||Outflow||
> |:-:|:-:|:-:|:-:|:-:|:-:|:-:|:-:|
> |||CPC|NMAE|NRMSE|CPC|NMAE|NRMSE|
> |CHI|TCA|0.585|0.205|0.307|0.581|0.209|0.310|
> ||MMD|0.638|0.194|0.280|0.626|0.200|0.289|
> ||ADV|0.616|0.200|0.304|0.613|0.204|0.307|
> ||CRAFT|**0.785**|**0.140**|**0.216**|**0.786**|**0.140**|**0.216**|
> |DC|TCA|0.659|0.229|0.315|0.658|0.231|0.317|
> ||MMD|0.696|0.236|0.301|0.708|0.237|0.295|
> ||ADV|0.713|0.237|0.287|0.703|0.249|0.302|
> ||CRAFT|**0.815**|**0.158**|**0.240**|**0.816**|**0.159**|**0.240**|
> |Toronto|TCA|0.692|0.274|0.359|0.688|0.280|0.360|
> ||MMD|0.631|0.261|0.356|0.640|0.257|0.349|
> ||ADV|0.724|0.238|0.299|0.716|0.243|0.308|
> ||CRAFT|**0.804**|**0.178**|**0.267**|**0.804**|**0.179**|**0.268**|
> |NYC|TCA|0.677|0.202|0.279|0.679|0.204|0.279|
> ||MMD|0.628|0.218|0.276|0.604|0.256|0.318|
> ||ADV|0.617|0.224|0.284|0.594|0.268|0.329|
> ||CRAFT|**0.782**|**0.103**|**0.170**|**0.786**|**0.102**|**0.165**|
>
> Compared to the original results in our paper, incorporating transfer learning indeed improves the baselines’ cross-city transfer performance, allowing them to be more task-specific. However, CRAFT still consistently retains SOTA performance due to the proposed RCA module, which provides selected priors from source cities.
>
> # Q2: Time complexity and efficiency
>
> |Method|Infer time(s)|Infer speed(samples/sec)|Training time(s)|
> |:-:|:-:|:-:|:-:|
> |GMEL|23.84|1701.63|5639|
> |CVAE|14.20|2883.65|2307|
> |CGAN|10.75|3775.23|12144|
> |DFG|46.54|871.63|36077|
> |KSTDiff|18003.47|2.13|73006|
> |Diffwave|420.83|96.40|3778|
> |DiT|1923.11|21.10|12430|
> |DDPM|1925.93|21.06|13024|
> |CRAFT-DDPM|1926.43|21.06|13492|
> |CRAFT-DDIM|21.08|1924.65|13492|
>
> In above table, we report both inference and training times for CRAFT and the baselines. CRAFT includes two variants: CRAFT-DDPM and CRAFT-DDIM. As expected, diffusion-based models are generally slower in both training and inference than single-shot generative models. CRAFT-DDPM achieves comparable efficiency to other diffusion-based models such as DiT but remains slower than single-shot models like GMEL. CRAFT-DDIM improves the sampling strategy of the diffusion backbone in the inference stage, achieving efficiency comparable to, or even surpassing, some single-shot models (e.g., DFG).
>
> |Target City|Method||Inflow|||Outflow||
> |:-:|:-:|:-:|:-:|:-:|:-:|:-:|:-:|
> |||CPC|NMAE|NRMSE|CPC|NMAE|NRMSE|
> |CHI|CRAFT-DDPM|0.785|0.140|0.216|0.786|0.140|0.216|
> ||CRAFT-DDIM|0.778|0.146|0.225|0.761|0.151|0.236|
> |DC|CRAFT-DDPM|0.815|0.158|0.240|0.816|0.159|0.240|
> ||CRAFT-DDIM|0.791|0.172|0.253|0.788|0.176|0.257|
> |Toronto|CRAFT-DDPM|0.804|0.178|0.267|0.804|0.179|0.268|
> ||CRAFT-DDIM|0.803|0.181|0.293|0.801|0.182|0.295|
> |NYC|CRAFT-DDPM|0.782|0.103|0.170|0.786|0.102|0.165|
> ||CRAFT-DDIM|0.778|0.196|0.186|0.734|0.105|0.169|
>
> In above table, we compare the performance of CRAFT-DDPM and CRAFT-DDIM. CRAFT-DDIM achieves performance close to CRAFT-DDPM and outperforms all other baselines. Therefore, for optimal performance, CRAFT-DDPM is preferred, while CRAFT-DDIM offers a better trade-off between performance and efficiency.
>
> # Q3: Can our model expand to traffic4cast?
>
> Due to limited time and computational resources, we selected Moscow and Berlin from the Traffic4cast[9] dataset to evaluate CRAFT.
> Compared to our initial datasets in the paper, Traffic4cast poses two main challenges:
>
> - finer-grained and more numerous regions result in a larger data scale that makes training more difficult
>
> - greater inter-city variation and lower similarity between regions across cities increase the risk of introducing irrelevant noise
>
> To address these issues, we made two adaptations:
>
> - we aggregated regions at the input stage to reduce region count, and used a linear layer at the output to map predictions back to the original resolution
>
> - we halved the retrieval numbers of RCA to reduce the possibility of invovling irrelevant noise. Results are in below table
>
> |Target City|Method||Inflow|||Outflow||
> |:-:|:-:|:-:|:-:|:-:|:-:|:-:|:-:|
> |||CPC|NMAE|NRMSE|CPC|NMAE|NRMSE|
> |Berlin|GMEL|0.500|0.442|0.501|0.487|0.470|0.528|
> ||DiT|0.436|0.353|0.444|0.453|0.494|0.571|
> ||Diffwave|0.424|0.562|0.662|0.386|0.710|0.768|
> ||DDPM|0.388|0.333|0.395|0.389|0.342|0.414|
> ||CRAFT|**0.537**|**0.312**|**0.381**|**0.537**|**0.313**|**0.381**|
> |Moscow|GMEL|0.182|0.357|0.431|0.254|0.345|0.418|
> ||DiT|0.566|0.403|0.480|0.500|0.320|0.396|
> ||Diffwave|0.170|0.397|0.469|0.486|0.530|0.594|
> ||DDPM|0.338|0.334|0.409|0.334|0.334|0.409|
> ||CRAFT|**0.609**|**0.239**|**0.315**|**0.607**|**0.240**|**0.316**|
>
> The experimental results demonstrate that CRAFT not only generalizes to the traffic4cast dataset but also shows bigger advantages over baselines than our original datasets, highlighting CRAFT’s strong generalization capability. We sincerely thanks for your suggestion and will include these results in the revised version. And we will also considering increasing the number of source cities in our future work.
>
> [9] NeurIPS traffic4cast 2021

---

> ### Author Response · Authors · 2025-08-08
>
> Dear Reviewers 6r4v,
>
> We sincerely thank you for your time, thoughtful evaluation, and constructive comments on our work. We truly appreciate your recognition of our model. We hope that our responses can address your concerns, which we summarize as follows:
>
> - To Weakness 1 and Question1: We conducted a more comprehensive review of flow generation and analyzed the unique challenges of cross-city scenarios, including the absence of task-specific baselines. These revisions help clarify that our contribution lies in serving as an initial probe into cross-city flow generation.
>
> - To Weakness 2 and Question1: We clarified how our model relates to existing transfer learning approaches and integrated three representative methods to strengthen the existing baselines, along with corresponding comparison experiments.
>
> - To Question 2: We conducted experiments on time complexity and efficiency. Specifically, we provide a detailed comparison of training and inference costs between our model and the baselines across all datasets.
>
> - To Question 3: We expand our model to Traffic4Cast. Specifically, we select Berlin and Moscow from the traffic4cast to evaluate the generalization ability of our model. Additionally, we discuss the unique challenges posed by Traffic4cast and describe the targeted adaptations we have made to address them.
>
> Thank you again for your valuable feedback. We truly appreciate the opportunity to clarify the motivations and contributions of our work.  As the rebuttal phase comes to an end, we always welcome any additional comments or questions you may have. Please feel free to share them with us, and we are always glad to engage in continued discussion.
>
> We hope that our responses have helped clarify your concerns. We would kindly ask if, you find our responses reasonable and aligned with your expectations, could you possibly reconsider and adjust the final score. We would be deeply grateful for your reconsideration.
>
> Regardless, we always fully respect your final decision and sincerely appreciate the time and patience you have devoted to reviewing our paper.
>
> Sincerely,
>
> The Authors

---

### Official Review · Reviewer_oBWv · 2025-07-02

**Clarity:** 4
**Significance:** 4
**Originality:** 3
**Rating:** 6
**Confidence:** 5

**Summary:**

This paper proposes a novel conditional-augmented diffusion model framework for addressing cross-city transfer issues in traffic flow data generation. Specifically: Aiming at the domain shift problem caused by the geographical feature domain shift of different cities, the authors propose a geographic representation alignment training method to extract high-quality representations and establish cross-city correspondences between regions. To tackle the insufficiency of dynamic generation conditions, the authors design a retrieval-augmented method based on data-rich cities. They make full use of the abundant data from source cities to enhance the model's performance in target city. From the reader's perspective, the most appealing aspect of this paper is the model's cross-city zero-shot generation capability, which is a highly demanded ability in spatio-temporal data analysis.

**Questions:**

See Weaknesses.

**Ethical Concerns:**

["NO or VERY MINOR ethics concerns only"]

**Final Justification:**

My major concerns on data dependency and experimental settings have been well answered. And after reading other reviewers' comments and the authors rebuttals, I believe this paper is valuable for the trafffic flow generation community.

**Limitations:**

Yes

**Quality:**

4

**Strengths And Weaknesses:**

Strengths:

S1. The authors propose a novel framework of cross-city RAG and optimal transport-based representation alignment. For the problem of cross-city zero-shot generation, they effectively capture the similarity factors between cities, making full use of abundant data from source cities to enhance the model's generalization and scalability.

S2. The research on cross-city traffic flow data generation is critical for the spatio-temporal data field. Most existing studies focus on traffic state prediction, but deep learning-based paradigms rely heavily on massive historical data for training, failing to predict traffic states in cities or regions with insufficient data. How to transfer deep learning models to data-scarce cities remains a challenging problem, and this paper provides valuable insights.

S3. The writing is concise and clear, with a lucid elaboration on the cross-city migration problem and corresponding challenges, enabling readers to grasp key points easily. The experimental analysis is also comprehensive.

Weaknesses:

W1. According to the authors' description, the traffic flow generation problem does not require historical traffic flow data as input but relies on basic geographical information. It is recommended that the authors elaborate on the data dependencies of this method in the experiments or appendix, i.e., what minimum inputs researchers need to prepare to synthesize data using this method.

W2. Some writing details in the paper need improvement. The Preliminaries section defines a graph composed of regions, but the subsequent text does not explicitly explain how the model models the adjacency relationships of urban spaces. The basic features of urban regions used later (POI, population, and road networks) should also be explicitly defined in the Preliminaries section.

W3. The abbreviations of different city datasets in the experimental section should be unified (e.g., Table 1 and Table 2 use full dataset names, while Figure 7 uses abbreviations) and explained to ensure the paper's rigor and readability.

---

> ### Author Rebuttal · Authors · 2025-07-31
>
> Thanks so much for your detailed and constructive comments. Please see our below responses to your concerns.
>
> # W1: Data dependency
>
> The data required for our model includes static geographical feature data of both the source and target cities, as well as traffic flow data of the source city.
> Specifically, the following static geographical features must be provided to use our model, with their acquisition and processing methods as follows:
> + **POI features**: POIs of the source and target cities can be downloaded from OpenStreetMap and matched to the city's spatial units (in our experiments, grid regions are treated as spatial units). We then count the number of various types of POIs in each region and use the TF-IDF algorithm to calculate the score of each POI type in the region. Specifically, we treat POI categories as words, all POIs within a region as a document, and all POIs in the entire city as a corpus to compute TF-IDF scores. Finally, these scores are concatenated into a feature vector.
> + **Road network features**: All road segments in the source and target cities can be downloaded from OpenStreetMap and matched to the city regions. We then calculate the total mileage of each type of road in each region and concatenate these values into a feature vector.
> + **Population features**: Population statistics for the source and target cities are downloaded from WorldPop. The population count and population density in each region are calculated and included as part of the basic features of the region.
> These static geographical feature data all rely on public data sources, allowing researchers to easily obtain them for any city.
>
> In addition, training our model requires traffic flow data covering all regions of the source city (including inflow and outflow for each region at each time slice), with a recommended time span of over 3 months. Traffic flow data can typically be derived from trajectory data, OD data statistics, or specialized traffic flow datasets. Our model is not restricted to specific traffic modes—it can handle bicycle, car, or pedestrian flow—but the generated flow for the target city should match the type used for the source city.
>
> **Extended static geographic features**:
> The road network features used in the model do not include the topological structure of the road network. To verify whether topological structure information is helpful for traffic flow modeling, we later tried adding space syntax features (including degree, closeness and betweenness) that describe the road network topology to the geographic features and conducted the following experiments.
>
> |Target City|Method||Inflow|||Outflow||
> |:-:|:-:|:-:|:-:|:-:|:-:|:-:|:-:|
> |||CPC|NMAE|NRMSE|CPC|NMAE|NRMSE|
> |CHI|CRAFT|0.785|0.140|0.216|0.786|0.140|0.216|
> ||CRAFT+syntax|0.814|0.121|0.187|0.815|0.123|0.189|
> |DC|CRAFT|0.815|0.158|0.240|0.816|0.159|0.240|
> ||CRAFT+syntax|0.800|0.165|0.246|0.799|0.168|0.247|
> |Toronto|CRAFT|0.804|0.178|0.267|0.804|0.179|0.268|
> ||CRAFT+syntax|0.827|0.159|0.239|0.827|0.159|0.240|
> |NYC|CRAFT|0.782|0.103|0.170|0.786|0.102|0.165|
> ||CRAFT+syntax|0.779|0.124|0.194|0.775|0.113|0.188|
>
> We found that on the Chicago and Toronto datasets, adding road network topological features helps improve the generation effect of the target city. However, this phenomenon does not occur on the DC dataset with a simple road network structure and the NYC dataset with a complex road network structure. When using our model, researchers can also try to expand the space syntax features of the road network.
>
> # W2: Writing
>
> **Adjacency Construction:** Each region is modeled as a graph node. If two region rectangles share a common edge (i.e., are spatially adjacent), a bidirectional edge is established between them. The adjacency relationships among regions are represented as an adjacency matrix, which is encoded by a graph transformer and incorporated into the regional geographical representation. We will clarify this definition in the Preliminary section of the final version.
>
> **Geographical features of regions:** In the Preliminary section, we only mentioned that each graph node has three types of geographical features but did not specify the details of each type. We provide formal definitions of the three feature types in the appendix and we will add explanations of these geographical features in the Preliminary section in the revised version.
>
> # W3: Experimental settings
>
> Thank you for your suggestions, which are very helpful for us to improve the details and readability of the paper. Due to the length constraints of the paper, we used abbreviations for the names of cities (datasets) in some experimental figures and tables, but failed to explain them clearly.
>
> In Figure 7 and Appendix Figures 11-12, DC, NY (NYC), and TRT stand for Washington, New York, and Toronto respectively. To ensure rigor, we will follow your suggestion to standardize the abbreviations in the experimental setup section to avoid ambiguity.

---

> > ### Comment · Reviewer_oBWv · 2025-08-04
> >
> > Thanks for the authors' responses. My major concerns on data dependency and experimental settings have been well answered. And after reading other reviewers' comments and the authors rebuttals, I believe this paper is valuable for the trafffic flow generation community and I would like to increase my rating to Strong Accept.

---

### Note · Authors · 2025-08-12

We sincerely thank all reviewers for their time, patience, and constructive feedback.

We clarified our motivation, novelty, and contributions to reviewers, and we are grateful for their recognition. We thank all reviewers for their comments and positive evaluations on our study, which addresses an important yet underexplored problem: cross-city flow generation. As an initial prob at this task, our work proposes a novel framework that enables zero-shot flow generation in data-scarce scenarios.

During the rebuttal period, the reviewers provided thoughtful and constructive feedback. Below is a summary of the key points:

- **Generalizability:** We addressed the concerns of reviewers 6r4v, vAPG, and qp6z regarding the applicability of our model to a broader range of datasets. In particular, we conducted additional experiments on a sampled subset of the Traffic4cast dataset, which covers vehicle flow across diverse countries in Europe and Asia and exhibits substantial inter-city variation. The results show that our method consistently outperforms the baselines, demonstrating its generalizability.

- **Time Complexity and Efficiency:** In response to reviewer 6r4v，vAPG，qp6z‘s concerns regarding computational costs and efficiency, we conducted a comparative analysis between our model and the baselines. Specifically, we evaluated training time, inference time, memory usage, and scalability on larger-scale datasets.

- **Robustness:** To reviewer vAPG’s concern regarding whether our model can handle sudden changes in flow patterns, we constructed a dataset containing holiday-related flow surges and evaluated our method on it.
The results demonstrate that our model‘s good potential for handling flow surges.

- **Clarification of Model Designs:** Inspired by Reviewer 6r4v’s comments, we conducted a more comprehensive literature review to clarify the distinction between cross-city flow generation and flow prediction. We also explain our model design from the perspective of transfer learning. To address Reviewers oBWv and vAPG’s concerns about data dependency, we added a comparative experiment to better illustrate our geographic feature selection approach.

We are confident that these suggestions have significantly strengthened the manuscript. We commit to integrating the supplementary content from this rebuttal into the revised version. Finally, we would like to thank the reviewers once again for their time and patience.

Sincerely，

The Authors

---

### Decision · Program_Chairs · 2025-09-17

**Decision:**

Accept (poster)

**Comment:**

This paper introduces an improved diffusion-based approach for generating cross-city traffic flows. The key idea is to leverage retrieval-augmented generation (RAG) to incorporate historical data from source cities under similar conditions into the conditioning process.

The reviewers raised concerns regarding generalizability, time complexity, robustness, and model design. After the discussion, most of these concerns were satisfactorily addressed. With all final scores being positive, this submission represents a clear acceptance.